# Pupil diameter encodes the idiosyncratic, cognitive complexity of belief updating

Alexandre LS Filipowicz[1,2,3†]*, Christopher M Glaze[1,2,3†], Joseph W Kable[2,3], Joshua I Gold[1,3]

[1]Departments of Neurscience, University of Pennsylvania, Philadelphia, United States; [2]Departments of Psychology, University of Pennsylvania, Philadelphia, United States; [3]Departments of Computational Neuroscience Initiative, University of Pennsylvania, Philadelphia, United States

**Abstract** Pupils tend to dilate in response to surprising events, but it is not known whether these responses are primarily stimulus driven or instead reflect a more nuanced relationship between pupil-linked arousal systems and cognitive expectations. Using an auditory adaptive decision-making task, we show that evoked pupil diameter is more parsimoniously described as signaling violations of learned, top-down expectations than changes in low-level stimulus properties. We further show that both baseline and evoked pupil diameter is modulated by the degree to which individual subjects use these violations to update their subsequent expectations, as reflected in the complexity of their updating strategy. Together these results demonstrate a central role for idiosyncratic cognitive processing in how arousal systems respond to new inputs and, via our complexity-based analyses, offer a potential framework for understanding these effects in terms of both inference processes aimed to reduce belief uncertainty and more traditional notions of mental effort.

**\*For correspondence:**
alsfilip@pennmedicine.upenn.edu

[†]These authors contributed equally to this work

## Introduction

Pupil size changes primarily as a function of ambient light levels but also in the context of a host of other brain functions that often involve changes in arousal (*Joshi and Gold, 2020*; *Mathôt, 2018*). Several early studies ascribed arousal-related modulations of pupil size to the 'mental effort' required to, for example, solve an arithmetic problem or hold items in working memory (*Beatty, 1982*; *Mathôt, 2018*). This view emphasizes the idiosyncratic and cognitive nature of these pupil fluctuations but lacks a precise description of the computations being encoded. Pupil size is also sensitive to stimulus probabilities, typically increasing when unexpected stimuli are presented or expected stimuli are omitted (*Joshi et al., 2016*; *Murphy et al., 2014*; *Qiyang et al., 1985*). More recently, arousal-related pupil modulations have been shown to represent critical computations required to learn from, and adapt to, changes in our environment (*Krishnamurthy et al., 2017*; *Nassar et al., 2012*; *O'Reilly et al., 2013*; *Preuschoff et al., 2011*). This view emphasizes specific computations encoded in pupil size but lacks a clear link to idiosyncratic, cognitive processing. The goal of the present study was to reconcile these different views and synthesize a more comprehensive understanding of the role of pupil-linked arousal systems in cognition.

We build on studies of adaptive decision-making, in which past experiences are used flexibly to update beliefs about the world that guide decisions (*Filipowicz et al., 2016*; *Johnson-Laird, 2004*; *Tenenbaum et al., 2011*). In uncertain and changing environments, effective belief-updating strategies need to resist spurious noise while remaining flexible enough to adapt to real changes (*Behrens et al., 2007*; *Filipowicz et al., 2016*; *Nassar et al., 2010*; *O'Reilly, 2013*). Several key features of these adaptive processes are reflected robustly in dynamic pupil fluctuations. Rapid phasic increases in pupil size occur in response to surprising events that can drive changes in beliefs

(*Nassar et al., 2012*; *O'Reilly et al., 2013*; *Preuschoff et al., 2011*). Slower tonic increases in pupil size can encode increased belief uncertainty (*Krishnamurthy et al., 2017*; *Muller et al., 2019*; *Nassar et al., 2012*; *Urai et al., 2017*). These slower pupil changes are also those most closely linked with changes in mental effort (*Beatty, 1982*; *Mathôt, 2018*) and associated with individual differences in cognitive traits such as fluid intelligence (*Tsukahara et al., 2016*).

However, several key cognitive and idiosyncratic aspects of adaptive decision-making have yet to be linked clearly to pupil modulations. First, adaptive decision-making has been studied primarily in relatively stable environments, in which changes to stimulus properties occur infrequently and are therefore almost always 'surprising' (*Filipowicz et al., 2016*; *Krishnamurthy et al., 2017*; *Nassar et al., 2012*; *O'Reilly et al., 2013*). These environments thus conflate violations of top-down, cognitive expectations that govern surprise with more basic bottom-up sensory responses to stimulus changes that can also modulate pupil size (e.g., 'startle' or 'orienting' responses; *Bremner, 2009*; *Sara, 2009*). Second, adaptive decision-making requires identifying changes in properties of not just observable stimuli but also of latent (i.e., not directly observable) environmental variables that affect uncertainty (*Glaze et al., 2018*; *Glaze et al., 2015*; *Krishnamurthy et al., 2017*; *McGuire et al., 2014*). Third, computations related to surprise and uncertainty can be strongly idiosyncratic, reflecting the specific belief-updating strategy being used by the decision-maker to integrate information from the environment (*Faraji et al., 2018*; *Filipowicz et al., 2014*; *Filipowicz et al., 2018*).

The current study aimed to overcome these limitations by assessing pupil size under a range of cognitively demanding task conditions and with respect to individual differences in the belief-updating strategy used to solve the task (*Glaze et al., 2018*). A key feature of the task was the use of complementary conditions that had equally predictable stimulus sequences but different relationships between relevant bottom-up stimulus properties and top-down beliefs. We show that across these conditions, the most parsimonious account of task-driven pupil modulations was not in terms of bottom-up stimulus properties. Instead, pupil size reflected the trial-by-trial, top-down belief-updating process, with baseline pupil size encoding the strength of the current belief and stimulus-evoked changes in pupil size encoding violations of the current belief. Moreover, the task conditions that most strongly dissociated bottom-up from top-down processing gave rise to high individual variability in behavior and task-driven pupil modulations, both of which were related systematically to the complexity of each subject's belief-updating strategy. Together these results imply that task-driven changes in pupil-linked arousal can represent a window into the underlying cognitive operations that update and maintain task-relevant beliefs.

## Results

Seventy-eight subjects performed an auditory predictive-inference task that required them to predict on each trial which of two source locations (left or right) would generate an auditory tone (*Figure 1a*). The source generating the tones switched locations from right-to-left or left-to-right at unannounced points throughout the task. The per-trial probability of this source-location switch, or 'hazard rate' ($H$), was fixed at either a low ($H = 0.01$), intermediate ($H = 0.30$), or high ($H = 0.99$) value in blocks of trials that changed at unannounced points throughout the task (*Figure 1b*). To make this inference process more challenging, uncertainty was added by playing the tone from the opposite side of the generating source on 20% of trials ('opposite-location trials'). This design allowed us to dissociate top-down expectations from bottom-up stimulus properties (i.e., location switches) and required the subjects to adapt their expectations to the different hazard-rate conditions (*Glaze et al., 2018*; *Glaze et al., 2015*). Specifically, the low versus high hazard-rate conditions produced location switches that were predictably rare or common, respectively, and thus could in principle be used strategically to avoid overreacting to opposite-location trials. In contrast, the intermediate hazard-rate condition was less predictable, such that the best strategy was to predict a repeat of the previous location (similar to $H = 0.5$ for a perceptual-judgment task, which requires decisions based on only the most-recent stimulus; *Glaze et al., 2018*, *Glaze et al., 2015*).

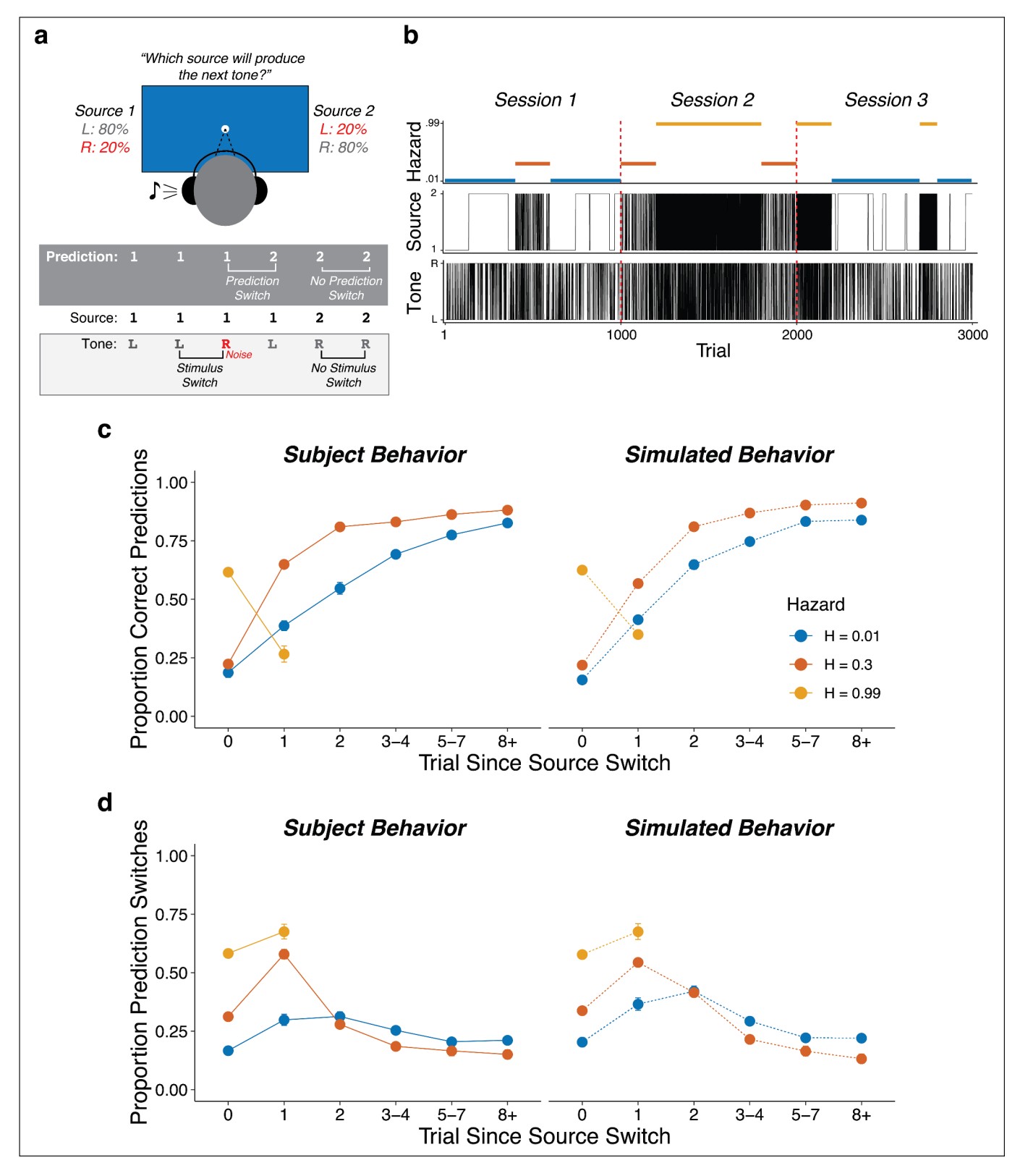

**Figure 1.** Auditory predictive-inference task and average performance. (**a**) Subjects predicted which of two sources would generate a tone on the next trial. After making the prediction, a tone was played in either the left or right ear. (**b**) Subjects completed up to three sessions of 1000 trials each. In each session, subjects were exposed to two hazard rates that would change at unannounced points throughout the session. (**c**) Proportion of subject

*Figure 1 continued on next page*

*Figure 1 continued*

and simulation correct predictions as a function of trials after a source-location switch, separated by hazard rate. (d) Proportion of subject and simulation prediction switches (i.e., changing their prediction from the previous trial) for trials after a source switch, separated by hazard rate. Simulated behavior was generated from the Bayesian learning model fit to data from individual subjects and applied to the same sequences observed by each subject. Error bars in c, d are ± SEM across real or simulated subjects.

## Pupil diameter is sensitive to adaptive, context-dependent expectations

On average, the subjects adjusted their behavior adaptively to different hazard rates. Across subjects, the proportion of correct predictions was above chance for all hazard-rate conditions, with the highest performance for low hazard rates (median [interquartile range, or IQR] proportion correct = 0.82 [0.79–0.85], 0.59 [0.56–0.61], and 0.63 [0.56–0.67] for low, intermediate, and high hazard rates, respectively; two-tailed signed-rank test for $H_0$: median = 0.5, all $ps < 4.2 \times 10^{-12}$; two-tailed paired rank-sum test for $H_0$: per-subject performance difference between hazard pairs = 0, all $ps \leq 0.0009$). More specifically, in the low hazard-rate condition, source-location switches from one side to the other were relatively rare and therefore unexpected. Accordingly, performance was low for trials on which the source location switched (because subjects reasonably assumed that the switch on that trial was from the 20% of opposite-location trials) but then gradually increased over several trials as the subject gathered more evidence that a real source-location switch had occurred. In the intermediate hazard-rate condition, although stability was still expected, source-location switches were slightly more expected. Accordingly, subjects had initially low performance for trials on which the source location switched, followed by a higher and more immediate propensity to switch predictions on the next trial. In the high hazard-rate condition, source-location switches were highly expected. Accordingly, subjects showed a relatively high prediction switch rate for trials on which the source location switched, with more errors on the rare non-switch trials (0.63 [0.56–0.67] proportion correct, p=$3.8 \times 10^{-11}$). These hazard-dependent behavioral patterns were roughly consistent with simulated behavior from a Bayesian learning model that used trial-by-trial observations to perform inference over both the current source and hazard rate (*Figure 1c,d*; *Glaze et al., 2018*).

To relate these adaptive behavioral patterns to pupil changes, we characterized how pupil diameter measured before ('baseline'), during ('evoked'), and after ('next baseline') stimulus presentation on each trial was modulated by task-relevant stimuli (i.e., presentation of the tone and whether or not it switched locations from the previous trial, which could result from either real source switches or opposite-location trials) and behaviors (i.e., whether or not the subject switched predictions from the previous trial). In general, the subjects' pupils dilated in response to tone presentation (*Figure 2—figure supplement 1*). The magnitude of this evoked pupil response was negatively correlated with the corresponding baseline pupil diameter (median [IQR] Spearman's *rho* comparing trial-by-trial baseline pupil diameter to baseline-subtracted evoked change per subject = −0.34 [−0.41 – −0.26], sign-rank test for $H_0$: median = 0, p=$1.1 \times 10^{-10}$), as has been reported previously (*de Gee et al., 2014*; *Gilzenrat et al., 2010*). To identify pupil modulations that were distinct from this relationship, we used two sets of linear regressions to relate task-relevant stimulus and behavioral features to: 1) baseline pupil diameter; and 2) changes in pupil diameter that were evoked by the tone presentation, with baseline pupil diameter subtracted out and using baseline pupil diameter as an additional regressor.

Both baseline pupil diameter and evoked pupil responses were affected by whether or not the tone switched locations from the previous trial, in a manner that depended strongly on the hazard-rate condition (i.e., the frequency with which the sound source switched locations). Specifically, in the low and intermediate hazard-rate conditions, the evoked changes in pupil diameter tended to be larger when the tone location switched than when it repeated from the previous trial. This effect persisted into the next baseline for the low, but not the intermediate, hazard-rate condition (*Figure 2*). Conversely, in the high hazard-rate condition, the evoked pupil responses were not modulated systematically by whether or not the sound location switched (*Figure 2c*), but baseline pupil diameter on the subsequent trial tended to be larger after the source location repeated than when it switched on the previous trial (*Figure 2b*).

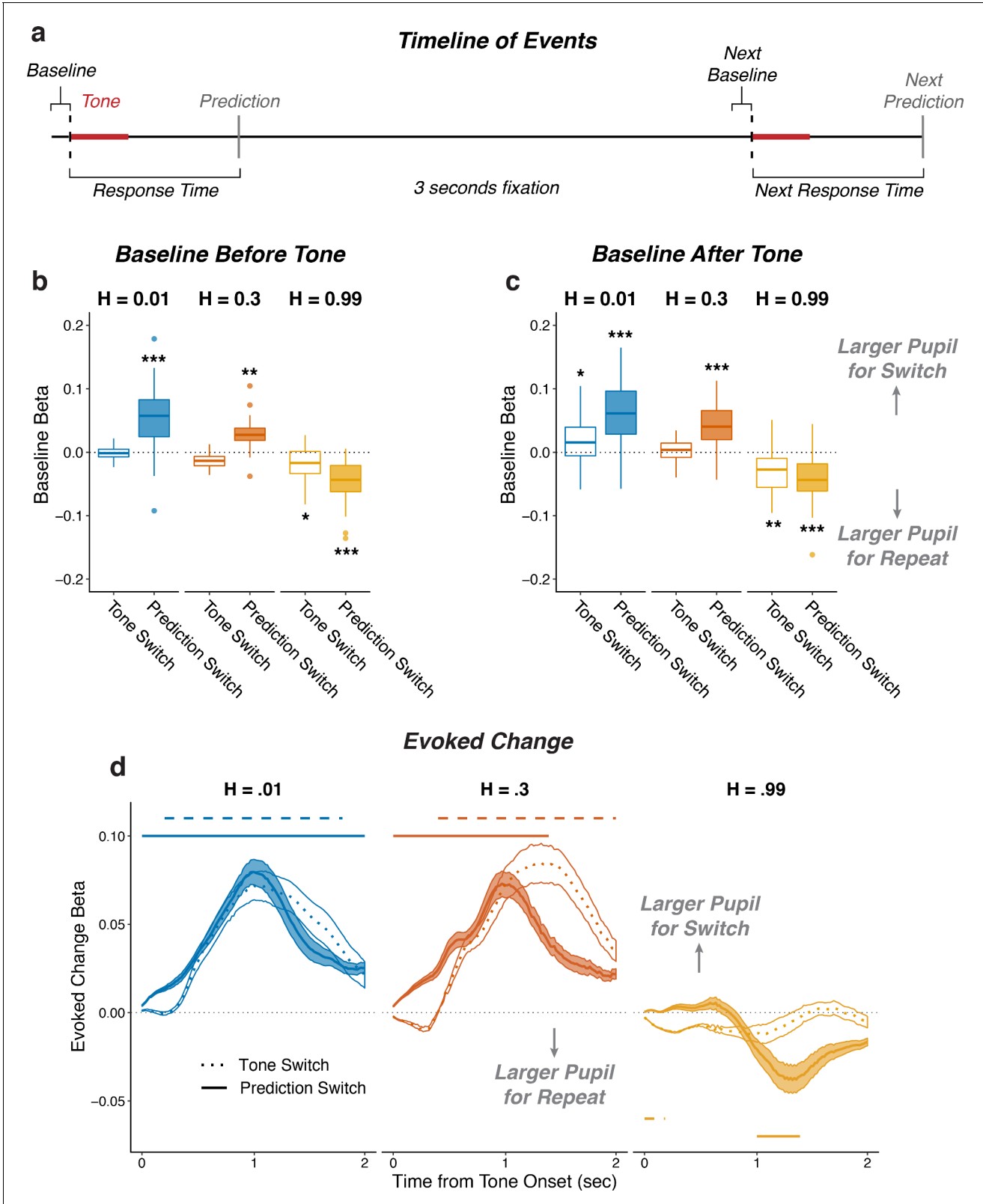

**Figure 2.** Pupil size encoded violations of expectations. (a) Sequence of tone and prediction events. (b–d) Each panel shows two-tailed linear mixed-effects beta weights with binary regressors corresponding to whether or not subjects switched their prediction and whether or not the tone switched sides on that trial, for: (b) baseline pupil diameter from the current trial, before the subject heard the tone and made the next prediction; (c) baseline pupil diameter from the next trial, after the subject heard the tone and made a prediction; and (d) changes in pupil diameter from baseline in a 2 s

*Figure 2 continued on next page*

*Figure 2 continued*

period after tone onset. In a and b, the regressions accounted for the evoked pupil change from the previous trial; boxplots show median, IQR, IQR x 1.5, and outliers; **: p<0.01, ***: p<0.001 estimated from linear regressions. In c, the regressions accounted for baseline pupil on the current trial along with the evoked pupil and baseline from the previous trial; lines and shading correspond to mean ± SEM across subjects, and straight lines above (for positive betas) and below (for negative betas) beta values indicate significant timepoints (FWE corrected p<0.05).
The online version of this article includes the following figure supplement(s) for figure 2:

**Figure supplement 1.** Average z-scored pupil diameter from tone onset across all subjects.
**Figure supplement 2.** Pupil encodes information about tone switches and prediction errors.

Both baseline pupil diameter and evoked pupil responses were similarly affected by whether or not the subject predicted that the sound source would switch locations from the previous trial, also in a manner that depended strongly on the hazard-rate condition. These effects reflected top-down, hazard-specific expectations that were distinct from the effects of switches in tone-source location, described above (*Figure 2*). Specifically, in the low and intermediate hazard-rate conditions, in which switches were unexpected, the baseline pupil diameter (measured prior to the subject hearing the tone and then making a prediction on that trial) was larger on trials in which the subject would then hear the tone and subsequently predict that the source location would switch versus repeat on the next trial (*Figure 2b*). The subsequent evoked pupil response showed an additional modulation in the same direction, corresponding to larger evoked dilations when the subject predicted a location switch versus repeat (*Figure 2c*). In contrast, in the high hazard-rate condition, in which switches were expected, baseline and evoked changes in pupil diameter tended to be larger when predictions repeated from the previous trial (*Figure 2a,c*). These effects also tended to persist into the next baseline pupil diameter (*Figure 2b*).

These stimulus- and behavior- driven effects on pupil were distinguishable from each other because the two were not always aligned; that is, tone and prediction switches did not always occur at the same time because of the generative noise, and because subjects made errors. Thus, another way of identifying whether and how pupil diameter was modulated in a context dependent manner was to characterize modulations by tone-location switch versus repeat, correct predictions versus errors, and their interactions (*Figure 2—figure supplement 2*). These analyses showed consistent increases in evoked pupil response after errors in all three hazard-rate conditions. Additionally, these pupil responses to errors were modulated by the hazard rate, such that pupils tended to dilate more to errors on tone-switch trials in the low hazard-rate condition, and to errors on tone-repeat trials in the high hazard-rate condition (measured as the tone-switch by error interaction in all three hazard conditions; *Figure 2—figure supplement 2*). Together these results show that pupil size was modulated by not just low-level stimulus properties (tone-location switches) but also context-specific predictions and their violations, corresponding to location switches when stability was expected and repetitions when instability was expected.

## Pupil diameter and response times encode adaptive, trial-by-trial belief updating

To better relate the task-dependent pupil modulations to the specific belief-updating strategies each individual subject used to derive hazard-specific predictions, we fit each subject's trial-by-trial sequence of behavioral choices (predictions) with a Bayesian learning model that performs inference over both the source location and hazard rate (*Glaze et al., 2018*). A key parameter in the model controls the width of the distribution of prior probabilities that subjects assign to different hazard rates, with wider prior distributions leading to model predictions that adapt more effectively to changes in hazard rate than narrower prior distributions (*Figure 3—figure supplement 1a*). The model combines this prior distribution with each new observation (i.e., on which side the tone is played) to update the strength of its belief about the current source location and hazard rate. Using the per-subject model fits, we quantified belief strength on each trial as the log-odds-ratio that source 1 versus 2 would generate the next tone (*Gold and Shadlen, 2001*). We quantified surprise on each trial as the negative log probability that the observed tone was predicted (*Filipowicz et al., 2018*; *Mars et al., 2008*; *O'Reilly et al., 2013*), which reflected the degree to which the new observation did not match the current belief about location (*Figure 3—figure supplement 1b*).

Baseline and evoked pupil diameter encoded belief strength and surprise, respectively, from this belief-updating process (*Figure 3a,b* shows results from linear regressions of pupil diameter that accounted for relationships between baseline and evoked pupil diameter as in *Figure 2*; *Figure 3— figure supplement 2* shows comparable results using a linear regression of the first temporal derivative of evoked changes in pupil diameter, which is thought to be less susceptible to baseline effects; *de Gee and Tsetsos, 2019*; *Joshi et al., 2016*; *Murphy et al., 2020*; *Zhao et al., 2019*). This encoding was similar for the three hazard-rate conditions, implying that the hazard-dependent effects of prediction and tone switches on pupil depicted in *Figure 2* reflected a common underlying relationship between pupil and belief updating. In particular, baseline pupil diameter was negatively correlated with the idiosyncratic model-derived belief strength (i.e., inverse uncertainty derived from subject-specific model fits), such that baseline pupil diameter was smaller when beliefs about the next source were more certain (*Figure 3a*). After the tone was heard, evoked changes in pupil diameter were closely associated with surprise, with more surprising stimuli evoking larger pupil dilations with a time course similar to the switch modulations (*Figure 3b*). Evoked change also reflected information about the prediction belief strength before having heard the tone, such that pupils dilated slightly less after tone onset if beliefs were very strong, regardless of surprise. Thus, on each trial, baseline and evoked changes in pupil diameter reflected top-down computations of uncertainty and surprise, respectively, in a manner that depended on the context in which the stimuli were experienced.

Similar effects were evident in the subjects' behavioral response times (RTs), which tended to be shorter on trials when beliefs were more certain and longer on trials following the occurrence of a surprising stimulus (*Figure 3c,d*). These RT effects testify to the behavioral relevance of these model-derived quantities but did not alone account for their relationships with pupil size: although there was a modest trial-to-trial correlation between the RT after hearing the tone and changes in pupil diameter on the same trial (median [IQR] Spearman's *rho* computed per subject = 0.11 [0.06– 0.18] for baseline pupil and 0.12 [0.07–0.18] for evoked pupil, sign-rank test for $H_0$: median = 0, all $p$s < $3.7 \times 10^{-7}$), baseline and evoked pupil changes showed the same strong modulations by belief strength and surprise, respectively, even when accounting for trial-by-trial RTs (Fig. *Figure 3—figure supplement 3*). Thus, task-dependent pupil modulations provided more information about the belief-updating process than simply how that process affected RT.

## Pupil dynamics reflect individual differences in belief-updating complexity

The preceding analyses showed group-level relationships between trial-by-trial belief updating dynamics and pupil diameter. However, the nature of this belief-updating process varied considerably across subjects. We reported previously that a major component of individual variability on this kind of task can be ascribed to differences in the complexity of each subject's belief-updating strategy (*Glaze et al., 2018*). Complexity measures the amount of past information a given belief-updating strategy encodes to predict future events, which captures the flexibility with which new observations are used to update the belief (*Bialek et al., 2001*; *Gilad-Bachrach et al., 2003*; *Myung et al., 2000*). Complex strategies can be helpful if the past features they encode are useful for prediction (e.g., tracking today's temperature, barometric pressure, and weather trends to predict tomorrow's weather) but can also lead to errors if the past features they encode are irrelevant (e.g., using today's European weather trends to predict tomorrow's weather in New York). Thus, a key signature of this form of complexity is a trade-off that is well known in statistics and machine learning between: 1) bias, or the (in) ability to flexibly update beliefs given new, relevant observations; and 2) variance, or the propensity to over-adjust beliefs given new, irrelevant (e.g., noisy) observations (*Bishop, 2006*). This bias-variance trade-off was evident in the present data set, such that subjects who tended to adjust their beliefs appropriately to real changes in hazard rate (low bias) also tended to have more variable choices even when the hazard rate remained fixed (high variance; *Figure 4—figure supplement 1*). Below we show that the corresponding individual differences in belief-updating complexity were reflected in certain features of task-driven pupil modulations.

We characterized individual differences in belief-updating complexity using an information-bottleneck analysis (*Palmer et al., 2015*; *Tishby et al., 2000*). Specifically, we quantified for each subject the mutual information between the sequences of task conditions on the previous trial (i.e., the sound location and hazard rate, which together we refer to as the past 'feature' *F*) and their current

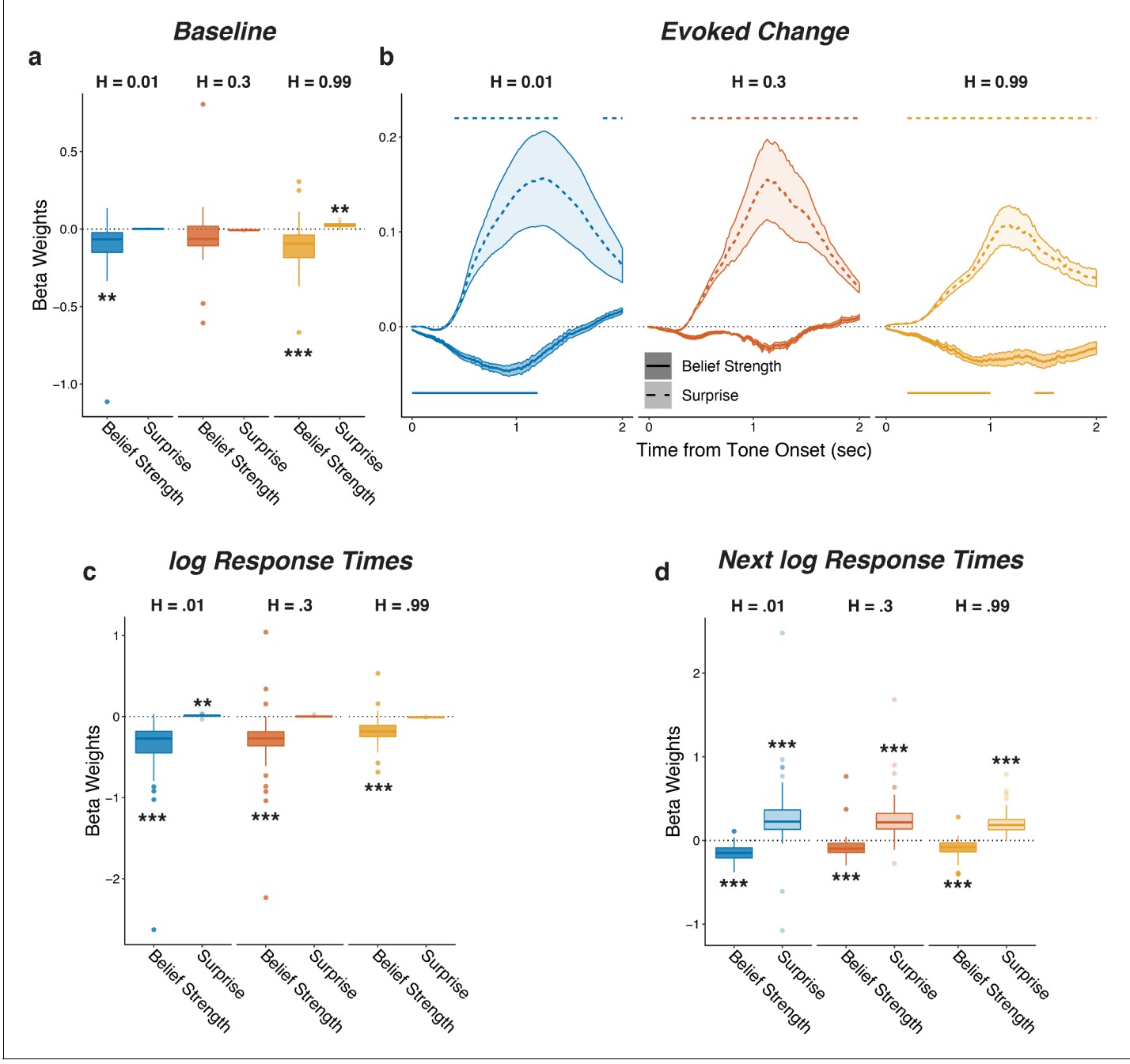

**Figure 3.** Pupil size encoded belief strength and surprise. Each panel shows two-tailed linear mixed-effects beta weights estimating the influence of subject absolute belief strength and surprise on: (a) baseline pupil diameter, after accounting for the evoked pupil change from the previous trial; (b) evoked change in pupil diameter, after accounting for baseline pupil on the current trial along with the baseline pupil and the evoked pupil from the previous trial; (c) log RT (i.e., the time taken to make a prediction before hearing the tone); and (d) log RT on the next trial. Note that surprise depended on the current belief strength that was present before tone onset, which likely explains the small modulations by surprise in the high-hazard baseline pupil and low-hazard prediction RT. In a, c, and d, boxplots show median, IQR, IQR x 1.5, and outliers; **: $p < 0.01$, ***: $p < 0.001$. In b, lines and shading correspond to mean ± SEM across subjects, and straight lines above (for positive betas) and below (for negative betas) beta values indicate significant timepoints (FWE corrected $p < 0.05$).

The online version of this article includes the following figure supplement(s) for figure 3:

**Figure supplement 1.** Bayesian model used to compute belief strength and surprise.

**Figure supplement 2.** Complimentary analysis to *Figure 3b* measuring the influence of belief strength and surprise on the first temporal derivative of evoked changes in pupil diameter.

**Figure supplement 3.** Pupils encode belief strength and surprise independently of RT.

predictions (*R*), after subtracting the information contained in the sound sequence (*Figure 4a,b*; *Chechik, 2005*; *Tishby et al., 2000*). This method provided a model-agnostic measure of the extent to which each subject's predictions relied on both stimulus and hazard-rate information, while also accounting for differences in the specific stimulus (tone-location) sequences that each subject experienced (*Glaze et al., 2018*). Unlike other measures of complexity that are based on a model's functional form (e.g., Bayesian model-selection methods; *Akaike, 1974*; *Myung et al., 2000*; *Schwarz, 1978*), this method provided a principled and data-driven measure of individual differences in belief-updating complexity that required no assumptions about the specific strategy each subject was using to perform the task (*Bialek et al., 2001*; *Chechik, 2005*; *Filipowicz et al., 2020*; *Tishby et al., 2000*). Consistent with definitions of model complexity in information theory and Bayesian model selection, more complex belief-updating strategies, such as those that explore larger hypothesis spaces, encode more information from the past to make predictions (*Bialek et al., 2001*; *Filipowicz et al., 2020*; *Myung et al., 2000*). As expected given this definition of complexity, subjects using more complex strategies tended to have behavioral patterns that depended more strongly on the hazard-rate condition (*Figure 4c*).

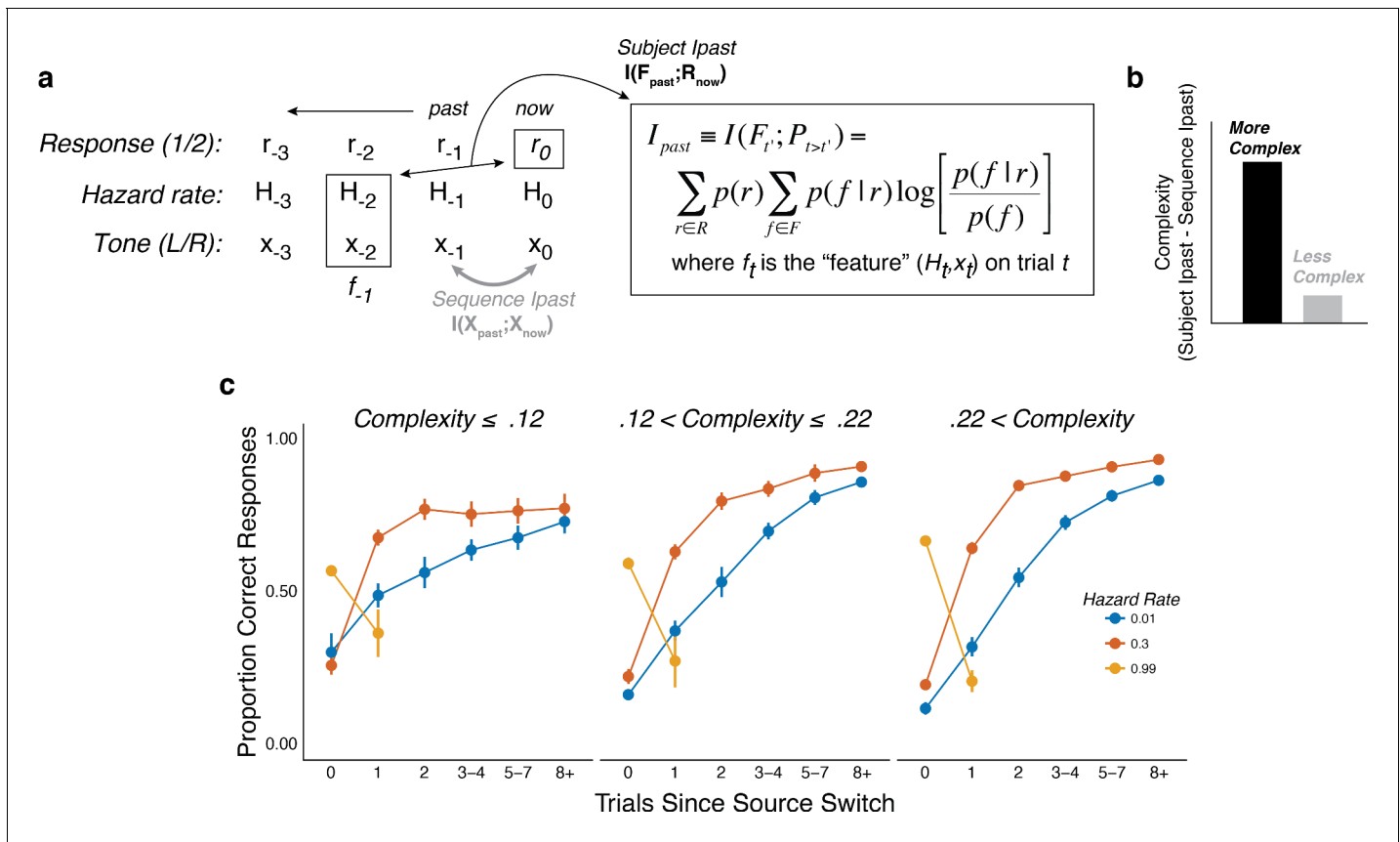

**Figure 4.** The subjects differed in belief-updating complexity. (a) The information-bottleneck measure of complexity, computed as the mutual information between past task features (the combination of the binary tone location, *x*, and hazard rate, *H*, from previous trials) and the current binary prediction, *r*. (b) To account for idiosyncratic differences in the sequences of tones that each subject heard, the mutual information between adjacent tones in the actual sequence ('Sequence *I_{past}*') was subtracted from their respective model complexity. The result corresponds to the degree to which subjects encoded hazard-related information in their predictions. (c) Average task performance plotted for groups of subjects split by complexity tercile (in bits). Subjects with more complex belief-updating strategies showed stronger hazard-modulated behavior.

The online version of this article includes the following figure supplement(s) for figure 4:

**Figure supplement 1.** Individual differences in performance reflected a bias-variance trade-off, similar to what we reported previously (*Glaze et al., 2018*).

**Figure supplement 2.** Parameter recovery for the: (a) adaptivity model and (b) Bayesian learning model.

These individual differences in belief-updating complexity were reflected in task-driven pupil modulations. Specifically, we used linear regression to relate changes in baseline pupil diameter from individual subjects to trial-by-trial estimates of belief strength that were computed using a particular, high-complexity learning model (i.e., the Bayesian learning model introduced above with a wide prior over hazard rate, which was not fit to subject data but rather used as reference strategy; *Figure 3—figure supplement 1a*) applied to each subject's task sequence. This relationship between baseline pupil fluctuations and complex belief strength became systematically stronger (more negatively correlated) for subjects who were using increasingly complex belief-updating strategies (*Figure 5*). This finding is consistent with our other findings, described above, that: 1) there was, on average, a negative correlation between trial-by-trial baseline pupil diameter and belief strength determined via fits of the Bayesian learning model to each subject's behavioral data (*Figure 3*); and 2) the subjects' belief-updating strategies varied considerably in complexity (*Figure 4a*). That is, given that each subject's pupil modulations reflected their own belief-updating strategy, and given that those strategies differed in complexity across subjects, it followed that their pupil modulations, measured with respect to a fixed referent strategy, reflected the complexity of their strategy. A similar relationship with complexity was not evident in the surprise-modulated evoked-pupil responses (*Figure 4b*), possibly reflecting relatively stronger bottom-up modulation in evoked versus baseline pupil. Together these results imply that certain features of trial-by-trial pupil dynamics can encode the idiosyncratic complexity of the cognitively driven belief-updating processes.

To better understand the trial-by-trial nature of these idiosyncratic, cognitively driven pupil modulations, we examined how complexity related to both behavior and changes in pupil diameter within each of the three hazard-rate conditions. Because our measure of complexity was determined by assessing patterns of choices across hazard-rate conditions, we first had to establish how it related to individual differences in performance within each hazard-rate condition. In general, subjects with

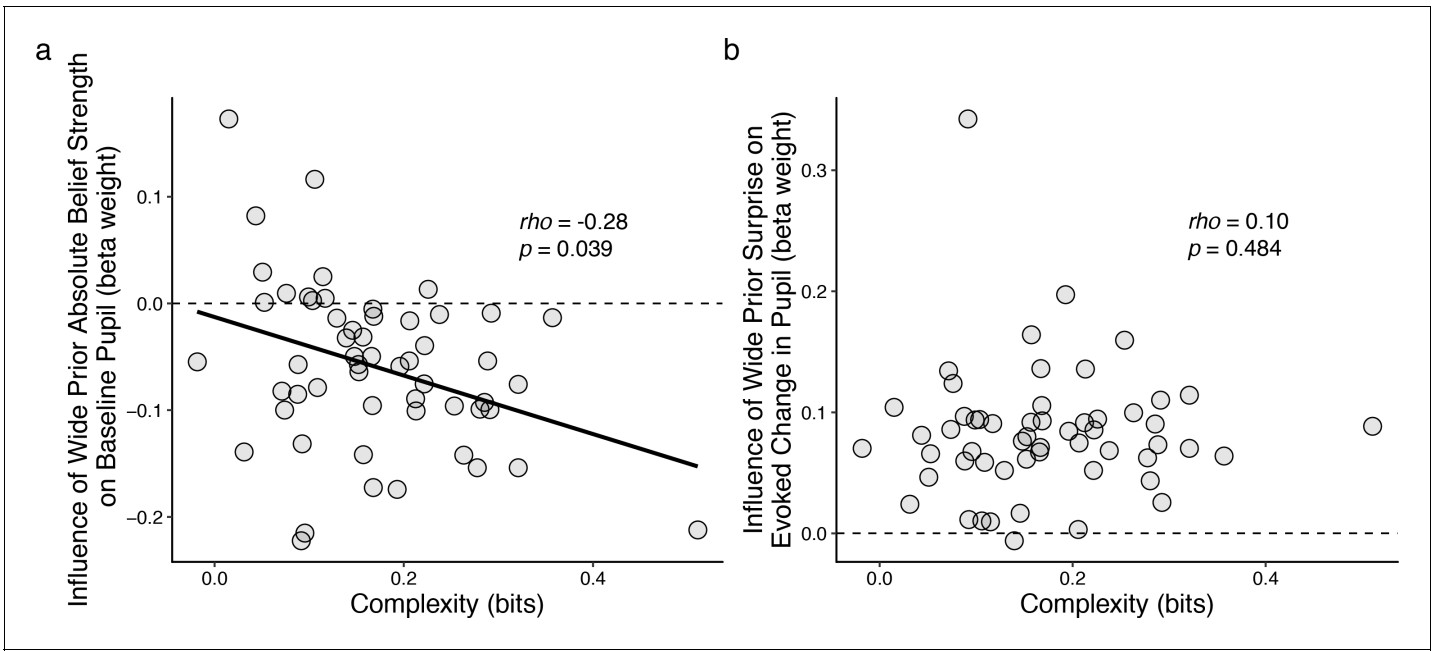

**Figure 5.** Individual differences in belief-updating complexity were reflected in across-hazard modulations of baseline, but not evoked, pupil diameter. (a) Strength of modulation of baseline pupil diameter by absolute belief strength from the complex learning model plotted as a function of the subject's belief-updating complexity. Subjects using simple strategies tended to have baseline pupil dynamics that were uncorrelated with the complex model's belief-updating process, whereas subjects using complex strategies tended to have baseline pupil dynamics that were anti-correlated (i.e., pupils were less dilated when the model predicted higher belief certainty) with the complex model's belief-updating process. The solid line is a linear fit. (b) Strength of modulation of peak evoked changes from baseline by surprise from the complex learning model plotted as a function of the subject's belief-updating complexity. Evoked changes in pupil diameter tended to be modulated positively by surprise in the complex model, regardless of the subject's belief-updating complexity. Points correspond to data from individual subjects. In both panels, the ordinates show the strength (computed via linear regression) of pupil modulations with respect to a particular, complex belief-updating strategy: the Bayesian model with a wide prior on hazard (*Figure 3—figure supplement 1*).

more complex strategies tended to make predictions that better matched the objective hazard rate (assessed by fitting the normative choice model to behavior with subjective hazard rate as a free parameter in hazard-specific blocks; *Glaze et al., 2015*; *Figure 6a*) and have higher overall accuracy (*Figure 6b*) in each of the three hazard-rate conditions.

Strikingly, however, both of these behavioral effects were most pronounced for the high hazard-rate condition. Several lines of evidence suggest that these effects reflected higher cognitive demands for the high hazard-rate condition, which as emphasized above (*Figures 2* and *3*) was the only condition that involved a conflict between bottom-up stimulus changes and top-down expectation violations, such that effective belief updating could not rely on stimulus changes alone to signal surprise. First, subjective hazard rates were most variable and most strongly mis-matched to objective values (dotted lines in *Figure 6a*) in the high hazard-rate condition (*F*-test comparing the variance of fits for each hazard rate condition: low versus high, p=2.2×10$^{-10}$; intermediate versus high,

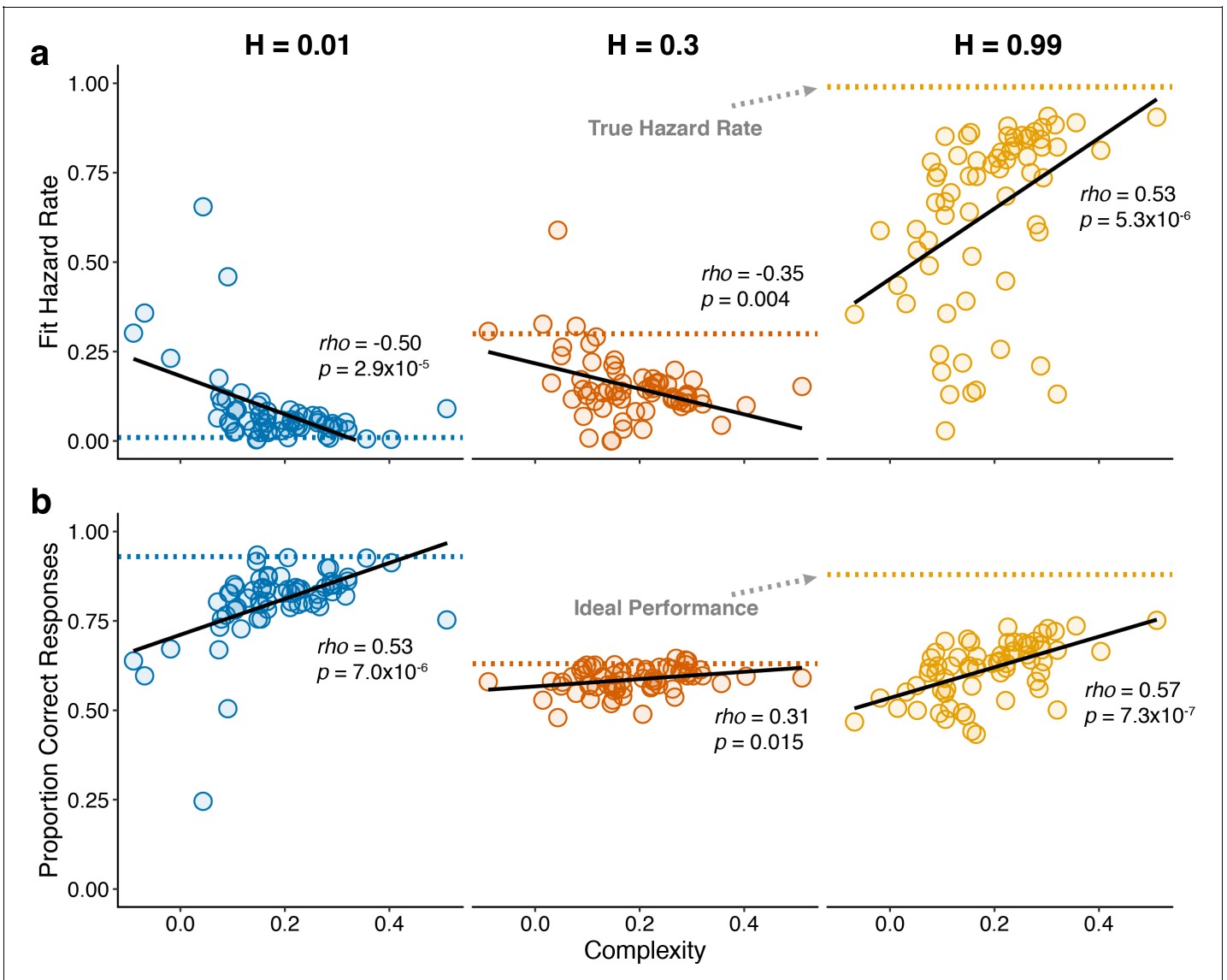

**Figure 6.** Individual differences in within-hazard performance reflected differences in belief-updating complexity. The two panels show complexity (abscissa) compared to: (a) subjective hazard rate determined from fits of the normative choice model (colored dotted lines correspond to objective hazard-rate values), and (b) proportion correct responses (colored dotted lines correspond to mean proportion correct responses from the Bayesian learning model with a wide prior over hazard rate). Points are data from individual subjects. Lines are linear fits. Note that subjective hazard rates and overall performance were relatively more variable and further from normative values for the high versus low and intermediate hazard-rate conditions.

$p=1.0\times10^{-13}$; low versus intermediate, $p=0.179$). Second, performance accuracy was furthest from the ceiling provided by a fully adaptive Bayesian learning model in the high hazard-rate condition (median [IQR] difference in accuracy compared with wide-prior model depicted in *Figure 3—figure supplement 1a* = 0.11 [0.08–0.15], 0.04 [0.03–0.07], and 0.29 [0.25–0.36] for the low, intermediate, and high hazard-rate conditions, respectively; two-tailed paired Wilcoxon tests comparing the difference in high versus low and intermediate hazards conditions, all $ps < 8.5\times10^{-10}$). Third, RTs were slowest in the high-hazard condition (median [IQR] RT for all trials and subjects = 478 [381–560], 507 [415–635], and 641 [513–790] ms for the low, intermediate, and high hazard-rate conditions, respectively; two-tailed paired sign-rank test for $H_0$: median difference = 0, all $ps < 0.0007$). Thus, despite the fact that the same level of accuracy could, in principle, be achieved in the high and low hazard-rate conditions, and that both conditions were more predictable than the intermediate hazard-rate condition, subjects had more difficulty learning in the high hazard-rate condition. Accordingly, only the best-performing subjects, who used the most complex belief-updating strategies, developed expectations for the highly unstable (high-hazard) condition that were as effective as the expectations developed by virtually all of the subjects for the highly stable (low-hazard) condition.

Consistent with these observations, individual differences in belief-updating complexity were reflected in pupil diameter in the high, but not the low or intermediate, hazard-rate condition. Specifically, subjects with more complex belief-updating strategies were able to learn much more appropriate expectations and thus perform closer to ceiling than subjects with less complex strategies in the high hazard-rate condition. For these subjects with complex strategies, tone switches versus repeats tended to more strongly reinforce their belief (i.e., reduce their uncertainty) that the source had switched, which corresponded to smaller baseline pupil diameter on the next trial (*Figure 7a*). Based on these more appropriate expectations, subsequent tone switches versus repeats tended to be less surprising, which corresponded to smaller evoked pupil responses (*Figure 7b*). In contrast, these relationships were not evident in the low and intermediate hazard-rate conditions, in which almost all subjects used strategies that led to near-ceiling levels of performance. These across-hazard differences have important implications for understanding the nature of cognitive demands that drive pupil modulations that we discuss below.

It was also notable that this relationship between belief-updating complexity and pupil diameter in the high hazard-rate condition was evident in both baseline and evoked pupil diameter. The relationship was stronger for baseline pupil diameter, which drove the complexity-dependent, across-hazard relationships between belief strength and baseline pupil shown in *Figure 5a*. In contrast, the slightly weaker relationship for evoked pupil did not produce a comparable complexity-dependent, across-hazard relationship between surprise and evoked pupil (*Figure 5b*), reflecting the fact that the evoked pupil responses were more strongly stimulus driven (*Figure 2*) and slightly less variable across subjects (*Figure 7*) than baseline pupil. Thus, under conditions in which individual differences in top-down belief-updating strategies had a substantial impact on behavior (i.e., in the more difficult high hazard-rate condition), an individual's strategy affected both how the pupil responded to task-relevant stimuli and then, to an even greater extent, modulations in the subsequent baseline pupil size that anticipated deployment of that strategy.

## Discussion

We examined relationships between pupil size and belief updating in human subjects performing an on-line predictive-inference task and report three primary findings. First, task-driven modulations of pupil size reflected context-dependent violations of learned expectations and not just bottom-up stimulus changes. Second, these cognitively driven changes in both baseline and evoked changes in pupil size could be parsimoniously understood as representing belief strength and surprise, respectively, associated with the subjects' trial-by-trial inferences about latent, task-relevant properties of the environment (in this case, a joint inference about the location of the source that would generate the next sound and the hazard rate that governed how often the location of the source switched from one side to the other). Third, these pupil dynamics reflected not just average belief-updating processes but also how they were implemented by individual subjects, further testifying to their behavioral and cognitive relevance. In particular, individual differences in belief-updating complexity were reflected in both baseline and evoked pupil responses under conditions in which the belief-

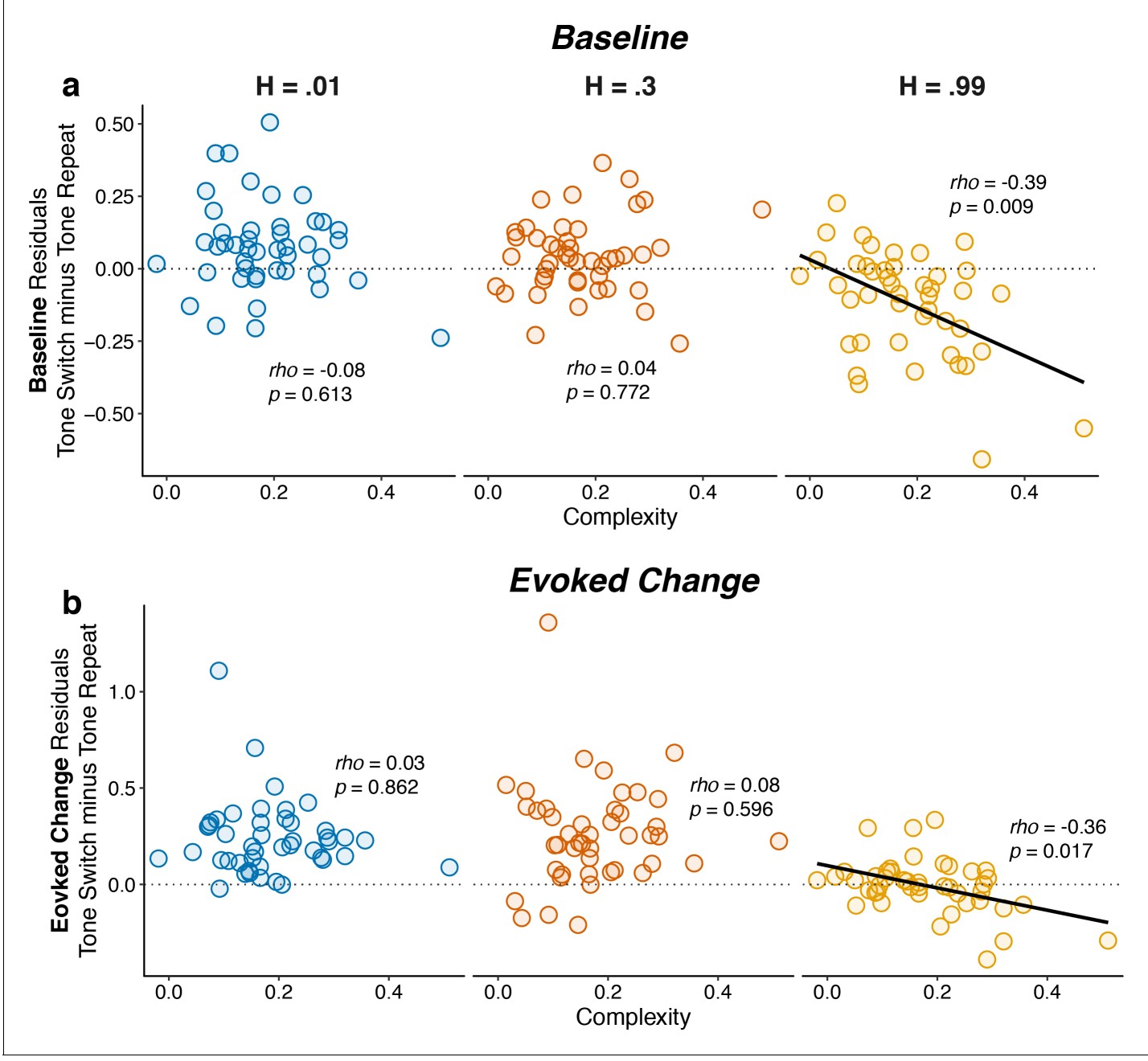

**Figure 7.** Individual differences in belief-updating complexity were reflected in within-hazard modulations of both baseline and evoked pupil diameter in the high hazard-rate condition that required a dissociation of top-down expectations from bottom-up stimulus changes. The two panels show (**a**) baseline and (**b**) evoked changes in pupil responses on tone switch versus repeat trials as a function of belief-updating complexity. Pupil measurements were computed as residuals from a linear model that accounted for nuisance variables including the baseline pupil diameter (for evoked changes) and the evoked change in pupil diameter from the previous trial (**Equation 14**). Points indicate data from individual subjects. Lines are linear fits for cases for which $H_0$: Spearman correlation coefficient (*rho*, indicated for each data set)=0, p<0.05.

updating process could not rely on bottom-up stimulus changes alone to identify surprising violations of expectations (the high hazard-rate condition).

These results help to clarify the nature of cognitive modulations of pupil size, including baseline or tonic values that have been associated with 'mental effort' (**Beatty, 1982**; **Mathôt, 2018**). According to our results, these modulations do not simply reflect: 1) objective task difficulty, because in our study objective difficulty was equivalent in the low and high hazard-rate conditions

that had substantially different pupil responses; 2) overall performance, because performance was similar in the intermediate and high hazard-rate conditions that also had substantially different pupil responses; or 3) RTs, which were only weakly related to pupil diameter on a trial-by-trial basis. Instead, our results suggest that tonic pupil-linked arousal is closely associated with a belief-updating process that aims to reduce uncertainty by interpreting new data according to multiple possible hypotheses (i.e., different hazard rates), which is consistent with a recent proposal that pupil-linked arousal encodes periods of information gain (*Zénon, 2019*). Our results extend this idea by showing that pupil modulations are also sensitive to the complexity of the belief-updating process, which may be related to traditional notions of cognitive load (*Beatty, 1982*; *Mathôt, 2018*). These ideas are related to previous findings showing that pupil dynamics can reflect uncertainty related to under-sampling (*Nassar et al., 2012*; *Krishnamurthy et al., 2017*) and to subjective estimates of environmental volatility (*Vincent et al., 2019*), which is a component of the belief-updating process that our subjects appeared to be using to solve the task.

A striking feature of the pupil-linked belief-updating process in our task was an asymmetry in how effectively it was applied to the different task conditions. As we reported previously, subjects tended to adapt incompletely to relatively predictable environments, including over-estimating low hazard rates and under-estimating high hazard rates (*Glaze et al., 2018*; *Glaze et al., 2015*; *Kim et al., 2017*). Here we showed that, despite the fact that those two hazard-rate conditions were equally predictable, these incomplete adaptations were asymmetric between the two conditions, with considerably worse and more variable performance in the high hazard-rate condition. This asymmetry was mirrored in pupil responses, with weaker overall modulations by surprise that varied systematically across subjects with belief-updating complexity in the high hazard-rate condition. These asymmetric modulations of pupil and behavior reflected the fact that in the low hazard-rate condition, belief updating could use congruent bottom-up and top-down information (i.e., stimulus changes were surprising), whereas in the high hazard-rate condition, belief updating was based on top-down processing that was incongruent with bottom-up information (i.e., stimulus repeats were surprising). Thus, the high hazard-rate condition promoted a greater sensitivity of both behavior and pupil dynamics to individual differences in top-down belief-updating strategies. It would be interesting to see if such an asymmetry occurs more generally when bottom-up and top-down information processing is and is not in conflict, including in the context of other proposed roles for pupil-linked arousal changes such as reductions in choice biases (*de Gee et al., 2017*).

Our findings also provide insights into the neural mechanisms that underlie adaptive and dynamic belief-updating processes. Increases in pupil size have been tied to higher levels of activation of the brainstem nucleus locus coeruleus (LC), which is the primary source of norepinephrine (NE) to the brain (*de Gee et al., 2017*; *Joshi et al., 2016*; *Liu et al., 2017*; *Murphy et al., 2014*; *Joshi and Gold, 2020*). Adaptive gain theory proposes that tonic release of NE occurs during periods of active exploration (*Aston-Jones and Cohen, 2005*; *Aston-Jones et al., 1999*). This idea is consistent with our finding of increased baseline pupil size under conditions of low belief (high uncertainty). Specifically, although physical exploration is absent in our task, there may be a form of 'mental exploration' that occurs when beliefs are weak and alternative explanations for the observed data are sought in terms of the hypothesis space of possible hazard rates (e.g., the prior distribution over hazard rates in our Bayesian learning model; *Collins and Koechlin, 2012*; *Glaze et al., 2018*; *Wilson et al., 2010*). Our results suggest that this covert process, like active exploration in adaptive gain theory, may involve increases in activation of the LC-NE system (and possibly the basal forebrain-acetylcholine neuromodulatory system, which has also been linked to pupil dynamics; *Reimer et al., 2016*) during periods of high uncertainty (*Yu and Dayan, 2005*).

Phasic, event-driven release of NE, possibly corresponding to evoked pupil changes in response to surprise as described in our and other studies (*Joshi et al., 2016*; *Nassar et al., 2012*; *O'Reilly et al., 2013*; *Preuschoff et al., 2011*), may play a complementary role in adaptive inference. Instead of promoting exploration, phasic NE release is thought to result in widespread network 'resets' that interrupt and initiate the reorganization of neural networks engaged during a task, possibly reducing previously learned biases and enhancing the relevance of new observations (*Bouret and Sara, 2005*; *de Gee et al., 2017*; *Sara, 2009*). Our results suggest that this kind of network reset can occur for not just an abrupt, surprising change in sensory input, but also an abrupt, surprising lack of change in sensory input.

Sensitivity to such different kinds of violations of expectations would require the LC and other relevant brain systems to have access to a flexible, complexity-dependent combination of bottom-up and top-down information. One possible source of such information is the anterior cingulate cortex (ACC; *Aston-Jones and Cohen, 2005*), which is reciprocally connected with the LC and has been proposed to guide executive control, allocating when and how executive resources are used during cognitively demanding tasks (*Filipowicz et al., 2016*; *Shenhav et al., 2013*; *Shenhav et al., 2016*; *Shenhav et al., 2017*). During adaptive decision-making, the ACC and the anterior insula are active when changes are detected and mental models are updated (*Behrens et al., 2007*; *Donoso et al., 2014*; *McGuire et al., 2014*; *O'Reilly et al., 2013*; *Sarafyazd and Jazayeri, 2019*; *Stöttinger et al., 2015*). Moreover, these kinds of signals have been linked directly with computations related to surprise and uncertainty (*McGuire et al., 2014*; *Nassar et al., 2019*). The ACC and anterior insula mediate activation of prefrontal cortical regions, such as the anterior and dorsolateral prefrontal cortex (*Bastin et al., 2017*; *Domenech and Koechlin, 2015*; *Ham et al., 2013*; *Menon and Uddin, 2010*; *Stöttinger et al., 2015*), that are involved in the prospective and counterfactual reasoning that are characteristic of the exploratory strategies humans use to update mental models (*Daw et al., 2006*; *Domenech and Koechlin, 2015*; *Donoso et al., 2014*; *Koechlin and Hyafil, 2007a*; *Koechlin and Summerfield, 2007b*; *Stöttinger et al., 2015*; *Zajkowski et al., 2017*). Our results suggest that increased activity in the LC, along with the network of brain regions with which it connects, could be more engaged in subjects with more complex mental models, guiding some of the idiosyncratic strategies observed between subjects.

In summary, the results of this study provide new insights into the cognitive signals that drive pupil diameter, including combinations of bottom-up and top-down processing that are used in idiosyncratic ways to update and maintain appropriate beliefs about an environment. These results can help guide future research to examine how dynamic modulation of physiological arousal influences the information processing trade-offs humans use when learning in dynamic and uncertain environments.

# Materials and methods

## Auditory predictive-inference task

Seventy-eight University of Pennsylvania undergraduates (49 male, 29 female) performed an auditory predictive-inference task that required them to predict the source location (left or right) of the subsequently presented auditory tone on each trial. On each trial, the subject made a prediction via left and right buttons on a game pad to indicate which source they thought would generate the next tone. Three seconds after each prediction, a 300 ms auditory tone (196 Hz) was played in either the left or right ear via headphones. The next trial then started with no inter-trial interval; thus, RTs were measured from the onset of the tone on the previous trial to the onset of the button press on the current trial. Subjects were encouraged to maintain fixation on a white fixation circle presented on a blue screen placed in front of them throughout the task but were allowed self-timed breaks by breaking fixation after making a prediction. Visual and auditory stimuli were generated using Snow-Dots stimulus presentation software in MATLAB (https://github.com/TheGoldLab/Lab-Matlab-Control; copy archived at https://github.com/elifesciences-publications/Lab_Matlab_Control; *Gold et al., 2020*).

Subjects were informed that the left source had an 80/20 probability of generating left/right tones, respectively, and that the right source had an 80/20 probability of generating right/left tones, respectively. Subjects were also informed that one source was generating tones on each trial, and that the sources would sometimes switch. Subjects were also informed that the rate of these switches, or 'hazard rate', would also switch, such that on some trials, the sources switched either infrequently ($H = 0.01$; low hazard rate), at an intermediate rate ($H = 0.3$; intermediate hazard rate), or very frequently ($H = 0.99$; high hazard rate). Subjects performed 1–3 sessions of 1000 trials each, with each session occurring on a separate day. Although all subjects were scheduled to participate in all three sessions, most completed only one ($n = 39$) or two ($n = 29$) sessions but were still included in the analyses (linear mixed-effects models described below were used to account for missed sessions). Two of the three hazard rates were included in each session, and the hazard rates themselves also switched infrequently throughout the session. Session order was counterbalanced

between subjects. The sequence of sound-source locations were different across the three sessions for each subject but were the same across subjects. The exact sequences of tones, sources, and hazard rates for each of the three sessions are displayed in *Figure 1b*.

Subjects received $10 USD per hour of participation with an additional performance bonus for accumulated correct predictions (total per session payout was $10–20 USD). The study protocol was reviewed and approved by the University of Pennsylvania's Institutional Review Board (IRB Protocol # 816727). All subjects provided written informed consent prior to participating in the study.

## Measuring pupil diameter

Of the 78 total subjects, 55 had their pupils recorded while performing the task. Pupils were recorded using either a Tobii T60-XL sampling both eyes at 60 Hz ($n = 45$), or an SR Research Eyelink 1000+ sampling the right eye at 1000 Hz ($n = 10$). Signals from the Eyelink were subsequently down sampled to 60 Hz to make them comparable to signals from the Tobii. Baseline pupil diameter was measured as the mean pupil diameter 0–100 ms prior to tone onset. Evoked changes were measured from baseline-subtracted pupil diameter during a 2 s window after tone onset.

## Pupil processing

We removed pupil samples corresponding to: 1) abnormally fast changes in pupil diameter (defined as ±3 SD from the median temporal derivative), such as around blink events, and 2) any additionally samples in which the eye position fell 10% of the screen diameter away from the fixation point. Eye movements in the x and y direction were regressed from the remaining samples and missing values were then linearly interpolated and filtered using a first-order Butterworth filter with a 4 Hz cutoff (*Browning et al., 2015*; *Krishnamurthy et al., 2017*; *Nassar et al., 2012*). Trials with >50% missing/ interpolated samples were omitted from further analysis (~9% of trials were omitted; *Browning et al., 2015*).

## Complexity measure

We used a data-driven approach to measure belief-updating complexity based on the principles of predictive information and the information bottleneck (*Bialek et al., 2001*; *Glaze et al., 2018*; *Tishby et al., 2000*). The information-bottleneck method seeks the smallest compression $Z$ of past observations that retain maximum predictability about the future. This compression represents a 'model' that encodes the past observations (*Chechik, 2005*; *Tishby et al., 2000*). The size of the compression ($I_{past}$), corresponding to the model's complexity, is measured (in bits) as the mutual information between the $Z$ and past observations $X_{past}$, $I_{past} = I(X_{past}; Z)$.

Belief-updating complexity was measured as the extent to which a subject's responses $R$ (a prediction of the left or right source) on the current trial $t$ encoded past stimulus features $F$ from past trials $t' < t$:

$$I_{past} \equiv I\left(F_{t' < t}; R_t\right) = \sum_{r \in R} \sum_{f \in F} \Pr(r, f) \log_2 \left[ \frac{\Pr(r, f)}{\Pr(r)\Pr(f)} \right] \tag{1}$$

The features $F$ consisted of all six possible combinations of tone side (left or right) and hazard rate (0.01, 0.3, or 0.99) on each trial. Information-theoretic quantities such as mutual information can be biased by small sampling errors. Although this influence was mitigated in part by the fact that the sequence of tones was held constant in each session for each subject, not all subjects participated in all sessions. We therefore corrected for these biases by subtracting the mutual information contained within the sequence of tones themselves, $I(tones_{t' < t}; tones_t)$, from each subject's $I_{past}$ (*Glaze et al., 2018*). The remaining information corresponds to the amount of information subjects encode about the task hazard rate, beyond that provided by the stimuli.

## Normative choice model

To determine each subject's subjective hazard rate corresponding to each objective hazard-rate condition, we fit their predictions to a normative choice model with hazard rate as a free parameter, as we have done previously (*Figure 6a*; *Glaze et al., 2018*; *Glaze et al., 2015*). For this task, the subject's goal was to predict which of two sources would generate a tone on the current trial. The

model characterizes the belief $L$ on trial $t$ as the log-posterior-odds that the next tone will be generated by the first or second source:

$$L_t = \psi_t(L_{t-1}, H_t) + LLR_t \tag{2}$$

where $\psi_t L_{t-1}$ indicates the log-prior-odds that either source will generate the next tone and $LLR_t L_{t-1}$ indicates the log evidence that the tone was generated by the first or second source. Given that there are only two states, we assigned positive log-prior-odds to the probability that a tone would be generated by the right source. The log-prior-odds $\psi L_{t-1}$ were computed using the log-posterior-odds from the previous trial $L_{t-1}$ and incorporating the current hazard rate $H_t$:

$$\psi_t(L_{t-1}, H_n) = L_{t-1} + \log\left[\frac{1 - H_t}{H_t} + \exp(-L_{t-1})\right] - \log\left[\frac{1 - H_t}{H_t} + \exp(L_{t-1})\right] \tag{3}$$

The posterior belief for a trial $t$ was then calculated as the log-prior-odds plus the evidence provided on each trial, calculated as the log-likelihood ratio that the current tone ($x_t$) was generated by the left or right source ($z_1$ and $z_2$ respectively)

$$LLR_t = \log\left[\frac{\Pr(x_t|z_1)}{\Pr(x_t|z_2)}\right] \tag{4}$$

Subjective estimates of each separate hazard rate (*Figure 6a*) were computed by fitting the model in *Equation 3* separately for each hazard-rate condition, keeping $H$ as a free parameter in each case (i.e., three free parameters, one for each value of $H$). The model was fit to choice behavior by writing the probability of choosing the right source as a logistic function of the log-prior-odds $\psi_t$ as the decision variable, given that the subject's responses were made before hearing a tone on each trial:

$$\Pr(\hat{c}_t) = 1/(1 + \exp[-\psi_t]) \tag{5}$$

Fitting was performed using gradient decent with multiple starting points to find values of the three free parameters, (three values of $H$ for each respective hazard rate) that minimize the cross entropy:

$$e = -\sum_t (1 - c_t)\log(1 - \pr(\hat{c}_t)) + c_t \log(\Pr(\hat{c}_t)) \tag{6}$$

We also fit a variant of this model to assess each subject's bias-variance trade-off; that is, the relationship between their adaptivity to changes in hazard rate and their choice variability (*Glaze et al., 2018*). This model used the same computations as from *Equations 2–4*, but calculated subjective estimates of the log hazard-rate odds on trial $t$ ($J_{t,subjective} = \log[H_{t,subjective}/(1 - H_{t,subjective})]$) as a linear function of the current, objective log-hazard rate odds ($J_{t,objective}$):

$$J_{t,subjective} = J_{default} + m_H \times J_{t,objective} \tag{7}$$

where $J_{default}$ indicates the intercept and $m_H$ indicates the slope of the regression, which describes the subject's adaptivity to changes in the task hazard rate. Choices were generated by adding a free parameter $v$ capturing noise in the decision variable (i.e., choice variability):

$$\Pr(\hat{c}_t) = 1/(1 + \exp[-\psi_t/v]) \tag{8}$$

Although a lapse term was previously included in this model (*Glaze et al., 2018*), adding this term did not improve our model fits and thus was not included here (BICs were lower for a model with choice variability and no lapse compared to a model with choice variability and lapse in 63 of 78 subjects; paired Wilcoxon signed-rank test comparing BICs: p=0.0004). Parameters from this winning 'adaptivity' model were highly recoverable (*Figure 4—figure supplement 2a*), as we reported previously (*Glaze et al., 2018*).

## Bayesian learning model

We used the same hierarchical Bayesian on-line inference algorithm used previously as a normative learning model for the current task (*Glaze et al., 2018*). This model predicts which of the two sources $z$ is most likely to generate a tone on the next trial. This model assumes that the likelihood that the sources switch is governed by a hazard rate ($H$), which it must learn, and that there is a fixed probability $K$ that $H$ changes from trial to trial.

For each subject, $H$ was assumed to be generated from a beta distribution $\mathrm{Pr}_0(H) = Beta(\alpha, \beta)$ with mean $\mu$ and precision $\phi$ parameters, such that $\alpha = \mu\phi$ and $\beta = (1 - \mu)\phi$. The goal was to actively infer which of the two possible sources $z$ (right or left) would generate the next tone $x$. Tones were $Bernoulli(p)$ distributed, with $p$=0.8. To infer the next tone, we sought a distribution over source-space for a given trial $t$, using all of the previously heard tones thus far $\mathrm{Pr}(z_t|x_{1:t-1})$:

$$\mathrm{Pr}(H_t, z_t|x_{1:t-1}, \mu, \phi, K, p) = \frac{\mathrm{Pr}(H_t, z_t, x_{1:t-1}|\mu, \phi, K, p)}{\int_0^1 dH_t \sum_{z_{t-1}} \mathrm{Pr}(H_t, z_t, x_{1:t-1}|\mu, \phi, K, p)} \tag{9}$$

The numerator can be re-written recursively as:

$$\mathrm{Pr}(H_t, z_t, x_{1:t-1}|, \mu, \phi, K, p) = \int_0^1 dH_{t-1} \mathrm{Pr}(H_t|H_t - 1, K) \\ \sum_{z_{t-1}} \mathrm{Pr}(z_t|z_{t-1}, H_t) \mathrm{Pr}(H_t, z_t, x_{1:t-1}|, \mu, \phi, K, p) \tag{10}$$

Where the transition probabilities in hazard-rate space are defined as:

$$\mathrm{Pr}(H_t|H_{t-1}, K) = \begin{cases} 1 - K + K\mathrm{Pr}_0(H_t) & H_t = H_{t-1} \\ K\mathrm{Pr}_0(h_t) & H_t \neq H_{t-1} \end{cases} \tag{11}$$

and the transition probabilities in source space are defined as:

$$\mathrm{Pr}(z_t|z_{t-1}, H_t) = \begin{cases} (1 - H_t) & z_t = z_{t-1} \\ H_t & z_t \neq z_{t-1} \end{cases} \tag{12}$$

The normative model's belief $L_t$ about the location of the source on trial $t$ after hearing a tone was computed as the log-posterior-odds of each state:

$$L_t = LLR_t + \psi_t \\ LLR_t = \log\left[\frac{\mathrm{Pr}(x_t|z_1)}{\mathrm{Pr}(x_t|z_2)}\right] \\ \psi_t = \log\left[\frac{\int_0^1 dH_t \int_0^1 H_{t-1} \mathrm{Pr}(H_t|H_{t-1}, K)}{\int_0^1 dH_t \int_0^1 H_{t-1} \mathrm{Pr}(H_t|H_{t-1}, K)} \frac{\sum_{z_{t-1}} \mathrm{Pr}(z_{1t}|z_{1:t-1}, H_n) \mathrm{Pr}(H_{t-1}, z_{t-1}, x_{1:t-1}|\mu, \phi, K, p)}{\sum_{z_{t-1}} \mathrm{Pr}(z_{2t}|z_{1:t-1}, H_n) \mathrm{Pr}(H_{t-1}, z_{t-1}, x_{1:t-1}|\mu, \phi, K, p)}\right] \tag{13}$$

This model was fit with four free parameters: $\mu$, $\phi$, $K$ governing the learning process and an additional choice variability parameter, $v$, identical to the one in *Equation 8*. The model was fit by minimizing the cross-entropy function in *Equation 6*. As with the adaptivity model, a lapse parameter was omitted, because adding this term did not improve our fits (BICs were lower for the choice variability only model for 65 of 78 subjects; paired signed-rank test comparing BICs: $p$=4.5x10$^{-5}$). Parameters from this model were highly recoverable (*Figure 4—figure supplement 2b*), as we reported previously (*Glaze et al., 2018*).

This Bayesian model was used to derive individual dynamic changes in belief strength and surprise for the pupil analyses in *Figure 3*. Belief strength was measured as the absolute value of the trial-by-trial log prior odds ($|\psi_t|$; *Murphy et al., 2020*), and surprise was computed using the information theoretic "surprisal", measured as $-\ln[\mathrm{Pr}(x_t|z_t, H_t)]$ (*Filipowicz et al., 2018*; *Mars et al., 2008*; *O'Reilly et al., 2013*).

The simulations used to compute normative surprise and belief strength in *Figure 5*, and benchmarks for accuracy in *Figure 6a*, were generated using a Bayesian model with a prior defined by

parameters $\mu = 0.5$ and $\phi = 2$, which weighs all hazard rates equally while performing inference (see *Figure 3—figure supplement 1*).

## Individual differences analyses

Individual differences in pupil responses were calculated using mixed-effects models implemented with the 'lme4' package (*Bates et al., 2015*) in the R statistical language (*R Development Core Team, 2019*). These models included both fixed-effects (e.g., tone switch and prediction switch variables) and nuisance regressors (*Krishnamurthy et al., 2017*):

$$Pupil = \beta_0 + \beta_{1-n} + \beta_{nuisance} + u_0 + u_{1-n} + u_{nuisance} \tag{14}$$

where $\beta_{1-n}$ indicates the $n$ fixed-effects regressors, $\beta_{nuisance}$ indicates analysis-specific nuisance regressors, and $u$ indicate random intercepts and random slopes for each fixed effects and nuisance regressor of interest. The fixed effects from these models were used to assess overall trends (e.g., influence of switches or computational variables). The individual subject slopes were used to assess individual differences in pupil dynamics. When measuring baseline pupil diameter, previous change in pupil diameter from baseline (i.e., peak pupil after tone onset on the previous trial minus previous baseline) was used as a nuisance regressor. When measuring evoked changes, baseline pupil diameter from the current trial, pupil change from the previous trial, and baseline pupil from the previous trial were added as nuisance regressors (*Krishnamurthy et al., 2017*).

Significance testing for the linear mixed effects models was done using the 'lmerTest' package (*Kuznetsova et al., 2017*), which computes $p$ values for $F$ and $t$ statistics using Satterthwaite's method for approximating degrees of freedom (*Fai and Cornelius, 1996*).

## Acknowledgements

The authors would like to thank Kenan Saleh, Mohammed Kabir, and Ian Jean for their assistance with data collection. Funded by NSF-NCS 1533623, R01 EB026945, and NIMH F32 MH117924. The funders had no role in the study design, data collection and analysis, decision to publish, or preparation of the manuscript.

## Additional information

### Competing interests

Joshua I Gold: Senior editor, *eLife*. The other authors declare that no competing interests exist.

### Funding

| Funder | Grant reference number | Author |
| --- | --- | --- |
| National Science Foundation | NSF-NCS 1533623 | Joseph W Kable Joshua I Gold |
| National Institute of Biomedical Imaging and Bioengineering | R01 EB026945 | Joshua I Gold |
| National Institute of Mental Health | F32 MH117924 | Alexandre LS Filipowicz |

The funders had no role in study design, data collection and interpretation, or the decision to submit the work for publication.

### Author contributions

Alexandre LS Filipowicz, Conceptualization, Data curation, Software, Formal analysis, Funding acquisition, Validation, Visualization, Methodology, Writing - original draft, Project administration, Writing - review and editing; Christopher M Glaze, Conceptualization, Resources, Data curation, Software, Formal analysis, Validation, Investigation, Visualization, Methodology, Project administration, Writing - review and editing; Joseph W Kable, Conceptualization, Supervision, Funding acquisition,

Methodology, Project administration, Writing - review and editing; Joshua I Gold, Conceptualization, Supervision, Funding acquisition, Visualization, Methodology, Project administration, Writing - review and editing

### Author ORCIDs
Alexandre LS Filipowicz 🔟 https://orcid.org/0000-0002-1311-386X
Joshua I Gold 🔟 http://orcid.org/0000-0002-6018-0483

### Ethics
Human subjects: The study protocol was reviewed and approved by the University of Pennsylvania's Institutional Review Board. All subjects provided written informed consent prior to participating in the study. (IRB Protocol # 816727).

### Decision letter and Author response
Decision letter https://doi.org/10.7554/eLife.57872.sa1
Author response https://doi.org/10.7554/eLife.57872.sa2

## Additional files

### Supplementary files
• Transparent reporting form

### Data availability
All data that support the findings in this article are available at https://osf.io/4ahkx/. All code used in the preparation of this article can be found at https://osf.io/4ahkx/ and https://github.com/TheGoldLab/Analysis_Filipowicz_Glaze_etal_Audio_2AFC (copy archived at https://github.com/elifesciences-publications/Analysis_Filipowicz_Glaze_etal_Audio_2AFC).

The following dataset was generated:

| Author(s) | Year | Dataset title | Dataset URL | Database and Identifier |
|---|---|---|---|---|
| Filipowicz AL, Glaze CM, Kable JW, Gold JI | 2020 | Analysis code and Data: Pupil diameter encodes the idiosyncratic, cognitive complexity of belief updating | https://osf.io/4ahkx/ | Open Science Framework, 4ahkx |

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
