## [Decision Letter]

**Acceptance summary:**

This paper presents a rigorous investigation of the relationship between pupil-linked arousal and dynamic belief updating in humans. The authors present the most conclusive evidence so far, that pupil-linked arousal reflects a high-level form of surprise in the belief updating process, conclusively ruling out low-level change in sensory input as a possible driver. They go on to show that individual differences in the complexity of the updating process.

**Decision letter after peer review:**

[Editors’ note: the authors submitted for reconsideration following the decision after peer review. What follows is the decision letter after the first round of review.]

Thank you for submitting your work entitled "Pupil diameter encodes the idiosyncratic, cognitive complexity of belief updating" for consideration by *eLife*. Your article has been reviewed by three peer reviewers, including Tobias H Donner as the Reviewing Editor and Reviewer #1, and the evaluation has been overseen by a Senior Editor. The following individual involved in review of your submission has agreed to reveal their identity: Sander Nieuwenhuis (Reviewer #2).

Our decision has been reached after consultation between the reviewers. Based on these discussions and the individual reviews below, we regret to inform you that your work will not be considered further for publication in *eLife*.

Reviewers agreed that the study was interesting and generally well done. However, reviewers #2 and #3 felt that this work was better suited for a more specialised journal than *eLife*. In the discussion leading up to this decision, reviewer #2 stated: The first half of the Results section mainly replicates already published findings (Nassar et al., 2012; Glaze et al., 2015), while the new findings in the second half of the Results section are probably hard to understand for the broad readership of *eLife*. I also found the first major conclusion, that evoked pupil-linked arousal is more sensitive to "high-level" surprise than to low-level changes in sensory stimulation rather unsurprising. In general, I thought that the science was great but the paper not very suited for *eLife*. Reviewer #3 made suggestions for reframing the paper in a way, they felt, would make it more accessible for a broad readership (focussing on general take-home messages rather than framing conclusions with respect to specific models). All reviewers also made a number of suggestions regarding the presentation and analyses that are detailed in the individual reviews below. We hope that these will be useful for you.

Reviewer #1:

This is timely work into the involvement of pupil-linked arousal systems in dynamic belief updating. The novelty of this study is twofold:

1) Demonstration that pupil diameter is sensitive to "high-level" surprise (deviations from expectations about state), as opposed to low-level changes in sensory stimulation. The comparison between high and low hazard rate contexts was necessary for this.

2) Linking these cognitive pupil diameter modulations to individual differences in internal model complexity. This point is conceptually extremely interesting; in practice I found their results in this part difficult to interpret.

In addition, there are some replications of effects reported in Glaze et al., 2018.

Overall, I feel this work merits publication at *eLife*, however, after some substantial revisions.

1) Quantification of pupil measures.

I suspect that the way the authors quantify baseline diameter and evoked pupil responses will induce strong (negative) correlations between both measures, as has been reported many times in the literature (Gilzenrath et al., 2010; de Gee et al., 2014). The source of this anti-correlation is not fully understand (it may be trivial or even artifactual), and it generally complicates the interpretation of their correlations with computational variables.

– The authors should report these correlations.

– I encourage them to replace their "evoked pupil change" with the 1st temporal derivative of the pupil signal in the post-tone interval; this measure has many advantages as a metric of phasic arousal: it is more temporally precise, it more specifically tracks norepinephrine (rather than ACh) transients, and, most critically, it tends to decorrelate measures of phasic arousal from baseline diameter measures of tonic arousal, see de Gee et al., 2019.

– For all tonic arousal effects it would be interesting to see effects for the previous baseline, not only the current and next baseline (this could show, for instance, if switches in response hand are preceded or succeeded by an increase of baseline arousal).

2) Split by choice variability.

Split into high vs low variability subjects seems ad hoc. No motivation is provided for doing so in Results, beyond this: "some participants had very high choice variability, and the interesting relationships reported here were present for low variability participants but not high variability participants".

If model complexity is really a useful construct that captures key features of how individuals perform a task like this, then shouldn't it encompass not just low variability participants, but high variability participants also?

3) Individual differences.

The pupil individual difference effects in Figure 6 are highly specific, examining the relationship between estimated model complexity and particular pupil effects within each H condition. If pupil-linked arousal plays a critical role in context-dependent belief updating, wouldn't one expect pupil changes across H conditions to correlate with changes in behaviour/belief updating across conditions? Such an analysis would get even closer to the claim that pupil-linked arousal is involved in adaptation of the inference process to different environmental contexts.

4) Interpretation of hazard rate condition differences.

In Discussion, authors speculate that relatively poorer performance in high H condition is due to a failure to engage arousal systems, which in turn produces less effective adaptation to that particular context. But the direction of causality could be reversed – participants might inherently underestimate the H in this context (perhaps due to strong priors for more stable environments encountered in everyday life), and that their level of surprise/arousal is actually appropriate given their subjective H. I suggest this possibility be acknowledged and given equal weight in the Discussion.

5) Is it possible that in Figure 1B, the dashed and dotted lines for H=.99 have been flipped?

Reviewer #2:

This paper reports a highly detailed analysis of the fundamental concepts that are reflected in pupil diameter. The study has a rather large sample size. The first part of the Results section presents findings that are a bit trivial and other findings that replicate the authors' previous work (mainly Nassar et al., 2012 and Glaze et al., 2015). But the paper also presents a number of novel analyses showing how the arousal system encodes complex context-dependent measures of uncertainty and belief updating. I am not able to judge the details of the complexity measure analyses and the "adaptivity model" (Materials and methods).

1) The paper aims to provide a unified framework for understanding pupil effects in terms of both inference processes and more traditional notions of mental effort and cognitive load. I encourage the authors to discuss how their framework accounts for the typical findings that gave rise to those traditional notions. In addition, could they briefly discuss the relationship between their framework and this paper from Karl Friston's group: Vincent et al., 2019 (https://www.ncbi.nlm.nih.gov/pubmed/31276488).

2) It would be good if the authors are more explicit in the Results section about which findings are "old" and where the new findings start.

3) Subsection “Pupil diameter is sensitive to adaptive, context-dependent expectations” and Figure 2C: "These effects persisted into the next baseline pupil, with larger pupil dilation after both stimulus and prediction switches on low and intermediate hazard-rate trials." Do these findings reflect real baseline shifts? Or are three seconds simply not enough for the pupil response to the previous tone stimulus to return to baseline?

Reviewer #3:

In this study Filipowicz and colleagues investigated the relationship between pupil dilation, perceptual surprise and belief updating in ~80 human participants. The used an auditory one-arm bandit task with variable hazard rate. Baseline pupil and phasic pupil response to stimuli were investigated. A number of mathematical models are proposed to account for within-and between subject variation in pupil response.

I was very interested to read this study which is close to my area of research. However, I was not convinced that it would command sufficient interest outside the immediate field to merit publication in *eLife*. The first reason for this assessment is that the study is a detailed investigation of an experimental effect (pupil dilation in response to learning conditions and surprise) rather than addressing a deeper hypothesis about the brain or behaviour per se. The second reason is that the bulk of the results are presented in relation to specific models and again I am not convinced that non-specialist readers would know how to generalize these (although this could possibly be addressed by a re-write that emphasises the take-home points). Thirdly, in some cases, if a study addressed a relatively non-innovative question but had a rare or technically excellent dataset I would still think it might be a good fit for *eLife*, but the data in this study would not be hard to collect for most cog neuro labs.

I have the following specific comments:

1) Figure 1 and related text

Subjects (and the model) seem to hugely under-estimate the Hazard rate in high H=0.99 condition and overestimate it in low H=0.33 condition (based on Figure 1D). How can this be explained?

2) Figure 2 and related text

It seems to me that this non-model-based analysis needs more unpacking.

– The pupil effects are not broken down into correct (prediction and stimulus = same location) and incorrect trials but I would expect a large difference in the phasic response between these trial types. This should interact with hazard rate as in low H=0.01 case, prediction switch trials should often be correct as they will follow multiple stimulus shifts. Could correct v incorrect be included in the regression model?

– The task gives quite a few rich comparisons that could be explored before getting stuck in to the modelling.

I would have liked to see more basic contrasts presented, eg:

– Compare the phasic pupil response for unexpected switches (in H=0.01) and unexpected repeats (in H=0.99) which nicely mirror each other.

– Compare baseline pupil on prediction-switch trials that follow a stimulus switch (presumably, mainly in the H=0.01 condition) and on prediction-switch trials that follow a stimulus repeat (presumably, mainly in the H=0.99 condition).

– Compare phasic pupil response following stimulus switches that will and will not be followed by a prediction switch.

These are results in themselves and would give the reader a better sense of the data without having to interpret in the light of models etc.

3) Figure 3 and related text

As for Figure 2, can surprise (based on model beliefs) be teased apart from correct vs incorrect (ie simply whether the prediction and stimulus were in the same location)?

4) Model complexity

I am struggling to understand how model complexity was defined exactly (Figure 4A). What is meant by the mutual information between the features of past trials and the current response – surely there must be some intervening policy that predicts the correct response based on these past features – what is it? Am I right in thinking that the model complexity captures both the extent of the integration kernel, or also so qualitative features of model complexity (e.g. another level in model hierarchy that tells you that you should alternate n the H=0.99 condition)? What aspects of the model complexity measure capture these two processes and can they be teased apart?

5) Figure 4 and related text

The relationship between three model based parameters in Figures 4C,D,E looks suspiciously good to me! Could this have arisen because these parameters to some extent capture similar features in the data, rather than separate but correlated cognitive processes? If the authors were to simulate agents who varied on only one or two of the parameters (choice variability/adaptivity/model complexity), to what extent would they be able to recover parameter fits that vary only in that one parameter?

6) Figure 5

Can the model based results be related back to raw behaviour? How should Figures 1C,D differ for subjects showing high and low model complexity?

7) Figure 7

Am I right in thinking the y-axis on this figure is mean pupil size (or mean phasic pupil response) as a subject-measure? How is this defined given that the raw pupil size on the eye tracker depends mainly on irrelevant factors like the distance from the camera to the eye and the size of the eye? I'm sure there is a sensible definition but I didn't manage to find it in the text. Further, might you not expect an inverted U (Yerkes-Dodson) relationship between pupil dilation and model complexity?

[Editors’ note: further revisions were suggested prior to acceptance, as described below.]

Thank you for submitting your article "Pupil diameter encodes the idiosyncratic, cognitive complexity of belief updating" for consideration by *eLife*. Your article has been reviewed by two peer reviewers, including Tobias H Donner as the Reviewing Editor and Reviewer #1, and the evaluation has been overseen by Timothy Behrens as the Senior Editor.

The reviewers have discussed the reviews with one another and the Reviewing Editor has drafted this decision to help you prepare a revised submission.

Summary:

This is a very rigorous study of the involvement of pupil-linked arousal systems in belief updating. The novelty of this work lies in two central points:

1) Phasic responses of pupil-linked arousal systems reflect a high-level form of surprise, as opposed to simply environmental change. (The high-hazard condition included here was critical to reach this conclusion.)

2) The model-free analyses of "complexity" pinpointed idiosyncratic way, in which individual participants address the cognitive demands of the inference task studied here.

Essential revisions:

You have addressed most of the reviewer concerns in a rigorous and comprehensive fashion. Yet, we felt, that the following conceptual points would deserve further attention in the presentation and discussion of your current findings.

1) Complexity measure and pupil. Please do more to explain/make intuitive your complexity measure. An example would be welcome. Everyone should be able to understand the most novel finding.

2) Pupil response to the “absence of change”. It has already been shown that the pupil responds to the unexpected absence of a stimulus (Qiyuan et al., 1985): There's nothing on the screen during the inter-trial interval and then, against expectations, no stimulus is presented. This is another example of "there is no sensory change but still a clear pupil response", indicating that the pupil responds to the violated expectation and not to bottom changes in sensory input.

3) Discussion of effort and load. In a new discussion paragraph, you now exclude a number of operationalizations of effort/load as an explanation for your findings. This is quite helpful. However, you end the Abstract mentioning that you will provide a unified framework for understanding effects of new inputs on arousal systems "in terms of both inference processes aimed to reduce belief uncertainty and more traditional notions of mental effort". We took that to mean that you will offer new interpretations of classical pupil findings, that have previously been explained in terms of effort or load. As far as we see, there is no such unified framework presented in the paper. We feel the last sentence of the Abstract should be toned down.

---

## [Author Response]

[Editors’ note: the authors resubmitted a revised version of the paper for consideration. What follows is the authors’ response to the first round of review.]

Reviewer #1:This is timely work into the involvement of pupil-linked arousal systems in dynamic belief updating. The novelty of this study is twofold:1) Demonstration that pupil diameter is sensitive to "high-level" surprise (deviations from expectations about state), as opposed to low-level changes in sensory stimulation. The comparison between high and low hazard rate contexts was necessary for this.2) Linking these cognitive pupil diameter modulations to individual differences in internal model complexity. This point is conceptually extremely interesting; in practice I found their results in this part difficult to interpret.In addition, there are some replications of effects reported in Glaze et al., 2018.Overall, I feel this work merits publication at eLife, however, after some substantial revisions.1) Quantification of pupil measures.I suspect that the way the authors quantify baseline diameter and evoked pupil responses will induce strong (negative) correlations between both measures, as has been reported many times in the literature (Gilzenrath et al., 2010; de Gee et al., 2014). The source of this anti-correlation is not fully understand (it may be trivial or even artifactual), and it generally complicates the interpretation of their correlations with computational variables.

We agree and apologize for the confusion. We do indeed find a strong negative correlation between baseline pupil diameter and our measure of baseline-subtracted evoked change. Therefore, all of our analyses of baseline-subtracted evoked change include baseline as a nuisance regressor. We have revised the Results and Materials and methods sections to make this point more clearly.

– The authors should report these correlations.

We now report the correlations between baseline and evoked changes in pupil in the Results section.

– I encourage them to replace their "evoked pupil change" with the 1st temporal derivative of the pupil signal in the post-tone interval; this measure has many advantages as a metric of phasic arousal: it is more temporally precise, it more specifically tracks norepinephrine (rather than ACh) transients, and, most critically, it tends to decorrelate measures of phasic arousal from baseline diameter measures of tonic arousal, see de Gee et al., 2019.

We now include a complimentary analysis of the derivative of pupil responses, which shows the same strong surprise effects as our analysis of evoked pupil (new Figure 3—figure supplement 2). We report in the Results section that this measure is also negatively correlated with baseline pupil diameter.

– For all tonic arousal effects it would be interesting to see effects for the previous baseline, not only the current and next baseline (this could show, for instance, if switches in response hand are preceded or succeeded by an increase of baseline arousal).

We again apologize for the confusion. The baseline pupil diameter analysis in Figure 2A corresponds to the analysis the reviewer is suggesting: baseline arousal is higher when participants make the less common prediction type (prediction switches in the low/intermediate hazard, and prediction repeats in the high hazard), *before* the tone is heard. This result implies that tonic arousal is higher when subjects are more uncertain and more likely to make the less common predictions. We have clarified this result and our interpretation in the text.

2) Split by choice variability.Split into high vs low variability subjects seems ad hoc. No motivation is provided for doing so in Results, beyond this: "some participants had very high choice variability, and the interesting relationships reported here were present for low variability participants but not high variability participants".If model complexity is really a useful construct that captures key features of how individuals perform a task like this, then shouldn't it encompass not just low variability participants, but high variability participants also?

We agree that this analysis was somewhat ad hoc and confusing and have removed it from the paper.

3) Individual differences.The pupil individual difference effects in Figure 6 are highly specific, examining the relationship between estimated model complexity and particular pupil effects within each H condition. If pupil-linked arousal plays a critical role in context-dependent belief updating, wouldn't one expect pupil changes across H conditions to correlate with changes in behaviour/belief updating across conditions? Such an analysis would get even closer to the claim that pupil-linked arousal is involved in adaptation of the inference process to different environmental contexts.

We have made extensive revisions to the text, analyses, and figures to expand upon and clarify this important point. These changes include:

A new figure (Figure 5) showing that belief-updating complexity is represented in baseline pupil diameter when considering all trials across hazard-rate conditions, as the reviewer suggests. As described in the text: “given that each subject’s pupil modulations reflected their own belief-updating strategy, and given that those strategies differed in complexity across subjects, it followed that their pupil modulations, measured with respect to a fixed referent strategy (i.e., from the Bayesian model using a wide prior over hazard rate), reflected the complexity of their strategy.”A more thorough description of why the within-hazard effects are specific to the high hazard condition. In brief, that is the only condition with a mismatch between bottom-up change and top-down surprise. Because of this mismatch, individual differences in topdown strategies play a much larger role in how subjects treat tone switch versus nonswitch trials in the high versus other hazard conditions. Consequently, it follows that only in the high-hazard condition do we see systematic, complexity-dependent differences in how the pupil responds to tones on switch versus non-switch trials.

4) Interpretation of hazard rate condition differences.In Discussion, authors speculate that relatively poorer performance in high H condition is due to a failure to engage arousal systems, which in turn produces less effective adaptation to that particular context. But the direction of causality could be reversed – participants might inherently underestimate the H in this context (perhaps due to strong priors for more stable environments encountered in everyday life), and that their level of surprise/arousal is actually appropriate given their subjective H. I suggest this possibility be acknowledged and given equal weight in the Discussion.

We agree and have substantially revised the Discussion accordingly.

5) Is it possible that in Figure 1B, the dashed and dotted lines for H=.99 have been flipped?

We have checked and all of the lines in this figure were correctly displayed.

Reviewer #2:This paper reports a highly detailed analysis of the fundamental concepts that are reflected in pupil diameter. The study has a rather large sample size. The first part of the Results section presents findings that are a bit trivial and other findings that replicate the authors' previous work (mainly Nassar et al., 2012 and Glaze et al., 2015). But the paper also presents a number of novel analyses showing how the arousal system encodes complex context-dependent measures of uncertainty and belief updating. I am not able to judge the details of the complexity measure analyses and the "adaptivity model" (Materials and methods).1) The paper aims to provide a unified framework for understanding pupil effects in terms of both inference processes and more traditional notions of mental effort and cognitive load. I encourage the authors to discuss how their framework accounts for the typical findings that gave rise to those traditional notions. In addition, could they briefly discuss the relationship between their framework and this paper from Karl Friston's group: Vincent et al., 2019 (https://www.ncbi.nlm.nih.gov/pubmed/31276488).

We thank the reviewer for this useful suggestion and have expanded on this point in the Discussion.

2) It would be good if the authors are more explicit in the Results section about which findings are "old" and where the new findings start.

We have substantially revised our presentation of the results, relegating most of the “old” findings to the supplemental materials (or removed them entirely) and focusing on just the novel findings.

3) Subsection “Pupil diameter is sensitive to adaptive, context-dependent expectations” and Figure 2C: "These effects persisted into the next baseline pupil, with larger pupil dilation after both stimulus and prediction switches on low and intermediate hazard-rate trials." Do these findings reflect real baseline shifts? Or are three seconds simply not enough for the pupil response to the previous tone stimulus to return to baseline?

We did not test longer inter-trial intervals and thus do not know how long these effects last. However, we emphasize that these baseline pupil modulations were measured after taking into account the strong positive correlation between baseline pupil and the previously evoked change in pupil (i.e., the previous evoked change was included as a nuisance regressor in the linear model). Thus, these effects were not simply the persistence of the evoked change. We have clarified these points in the figure legend.

Reviewer #3:In this study Filipowicz and colleagues investigated the relationship between pupil dilation, perceptual surprise and belief updating in ~80 human participants. The used an auditory one-arm bandit task with variable hazard rate. Baseline pupil and phasic pupil response to stimuli were investigated. A number of mathematical models are proposed to account for within-and between subject variation in pupil response.I was very interested to read this study which is close to my area of research. However, I was not convinced that it would command sufficient interest outside the immediate field to merit publication in eLife. The first reason for this assessment is that the study is a detailed investigation of an experimental effect (pupil dilation in response to learning conditions and surprise) rather than addressing a deeper hypothesis about the brain or behaviour per se. The second reason is that the bulk of the results are presented in relation to specific models and again I am not convinced that non-specialist readers would know how to generalize these (although this could possibly be addressed by a re-write that emphasises the take-home points). Thirdly, in some cases, if a study addressed a relatively non-innovative question but had a rare or technically excellent dataset I would still think it might be a good fit for eLife, but the data in this study would not be hard to collect for most cog neuro labs.I have the following specific comments:1) Figure 1 and related textSubjects (and the model) seem to hugely under-estimate the Hazard rate in high H=0.99 condition and overestimate it in low H=0.33 condition (based on Figure 1D). How can this be explained?

We thank the reviewer for bringing up this interesting and important point, which we have also reported for other tasks with similar statistical structures (Glaze et al., 2015; Kim et al., 2017). In our 2018 paper (Glaze et al) that focused on the bias-variance trade-off, we showed that this observation can be accounted for (providing a “how” if not necessarily a “why”) via the normative learning model we use in the present study, in terms of a prior over hazard rate that tends to be centered around 0.4–0.5 for most subjects. We have previously speculated (in Kim et al) that this value may be the default for many people because it might minimize computational costs (e.g., Drugowitsch et al., 2012; Shenhav et al., 2013). We are actively working on this question.

We now cover this point explicitly in the Discussion.

2) Figure 2 and related textIt seems to me that this non-model-based analysis needs more unpacking.– The pupil effects are not broken down into correct (prediction and stimulus = same location) and incorrect trials but I would expect a large difference in the phasic response between these trial types. This should interact with hazard rate as in low H=0.01 case, prediction switch trials should often be correct as they will follow multiple stimulus shifts. Could correct v incorrect be included in the regression model?– The task gives quite a few rich comparisons that could be explored before getting stuck in to the modelling.I would have liked to see more basic contrasts presented, eg:– Compare the phasic pupil response for unexpected switches (in H=0.01) and unexpected repeats (in H=0.99) which nicely mirror each other.– Compare baseline pupil on prediction-switch trials that follow a stimulus switch (presumably, mainly in the H=0.01 condition) and on prediction-switch trials that follow a stimulus repeat (presumably, mainly in the H=0.99 condition).– Compare phasic pupil response following stimulus switches that will and will not be followed by a prediction switch.These are results in themselves and would give the reader a better sense of the data without having to interpret in the light of models etc.

We appreciate the very useful suggestions and now include more extensive analyses and discussions of the basic task-driven pupil dynamics before considering the learning model, including results from a linear regression with tone switch/repeat, correct/error, and the interaction between the two as regressors (Figure 2—figure supplement 2).

3) Figure 3 and related textAs for Figure 2, can surprise (based on model beliefs) be teased apart from correct vs incorrect (ie simply whether the prediction and stimulus were in the same location)?

As noted above, we now show pupil modulations by correct versus errors (and their interaction with tone switch/repeat trials; Figure 2—figure supplement 2). These modulations are consistent with the surprise-driven modulations we show in Figure 3. Note that in this task there is no separate feedback about correct or error responses separate from the sound presentation on the next trial, so high surprise (a violation of the prediction) is very strongly correlated with errors.

4) Model complexityI am struggling to understand how model complexity was defined exactly (Figure 4A). What is meant by the mutual information between the features of past trials and the current response – surely there must be some intervening policy that predicts the correct response based on these past features – what is it? Am I right in thinking that the model complexity captures both the extent of the integration kernel, or also so qualitative features of model complexity (e.g. another level in model hierarchy that tells you that you should alternate n the H=0.99 condition)? What aspects of the model complexity measure capture these two processes and can they be teased apart?

We apologize for the confusion and have substantially revised our description of the complexity measure to improve clarity. In short, yes, this measure is based on the assumption that the subject is using a policy (we use the term “strategy” to be more general) that makes predictions based on those features. Critically, however, as we note in the text, this method “requires no assumptions about the specific strategy the subject is using to perform the task (Bialek, Nemenman and Tishby, 2001; Chechik et al., 2005; Filipowicz et al., 2020; Tishby et al., 2000).” That is, with this measure we capture the amount of information the policy encodes but nothing about its form (although note that in Glaze et al., 2018, we showed that in the context of this kind of task, this measure is closely related to the width of the prior in the normative learning model, which we refer to in the present manuscript).

5) Figure 4 and related textThe relationship between three model based parameters in Figures 4C,D,E looks suspiciously good to me! Could this have arisen because these parameters to some extent capture similar features in the data, rather than separate but correlated cognitive processes? If the authors were to simulate agents who varied on only one or two of the parameters (choice variability/adaptivity/model complexity), to what extent would they be able to recover parameter fits that vary only in that one parameter?

We have included parameter recovery analyses for both the adaptivity model and Bayesian updating model (Figure 4—figure supplement 1) to show that our parameters are highly recoverable, and the correlations between adaptivity and choice variability are not due to biases in the model specification (as we showed in Glaze et al., 2018).

6) Figure 5Can the model based results be related back to raw behaviour? How should Figures 1C,D differ for subjects showing high and low model complexity?

Subjects with more complex beliefs show stronger hazard-dependent behavior (i.e., they adapt better to changes in hazard rate). We have included a panel showing changes in subject behavior as a function of different complexity terciles (Figure 4B).

7) Figure 7Am I right in thinking the y-axis on this figure is mean pupil size (or mean phasic pupil response) as a subject-measure? How is this defined given that the raw pupil size on the eye tracker depends mainly on irrelevant factors like the distance from the camera to the eye and the size of the eye? I'm sure there is a sensible definition but I didn't manage to find it in the text. Further, might you not expect an inverted U (Yerkes-Dodson) relationship between pupil dilation and model complexity?

The goal of this figure was primarily to demonstrate that the effects we observed in Figure 6 were due to real differences between tone switch and non-switch trials rather than some idiosyncrasies in overall baseline or evoked changes. However, our experimental set up does not allow us to make specific claims about how pupil diameter differs as a function of complexity, particularly not without accounting for all of the factors the reviewer mentions (e.g., viewing distance, eye size, etc). We have therefore removed this figure and mention of these analyses from our manuscript.

[Editors’ note: what follows is the authors’ response to the second round of review.]

Essential revisions:You have addressed most of the reviewer concerns in a rigorous and comprehensive fashion. Yet, we felt, that the following conceptual points would deserve further attention in the presentation and discussion of your current findings.1) Complexity measure and pupil. Please do more to explain/make intuitive your complexity measure. An example would be welcome. Everyone should be able to understand the most novel finding.

We have included a more detailed explanation and example of our complexity measure in subsection “Pupil dynamics reflect individual differences in belief-updating complexity” of the revised manuscript.

2) Pupil response to the “absence of change”. It has already been shown that the pupil responds to the unexpected absence of a stimulus (Qiyuan et al., 1985): There's nothing on the screen during the inter-trial interval and then, against expectations, no stimulus is presented. This is another example of "there is no sensory change but still a clear pupil response", indicating that the pupil responds to the violated expectation and not to bottom changes in sensory input.

We have added a sentence to the Introduction to highlight this finding and have added this reference.

3) Discussion of effort and load. In a new discussion paragraph, you now exclude a number of operationalizations of effort/load as an explanation for your findings. This is quite helpful. However, you end the Abstract mentioning that you will provide a unified framework for understanding effects of new inputs on arousal systems "in terms of both inference processes aimed to reduce belief uncertainty and more traditional notions of mental effort". We took that to mean that you will offer new interpretations of classical pupil findings, that have previously been explained in terms of effort or load. As far as we see, there is no such unified framework presented in the paper. We feel the last sentence of the Abstract should be toned down.

We have revised the last sentence of the Abstract.